# Gene autoregulation by 3' UTR-derived bacterial small RNAs

**Mona Hoyos[1,2], Michaela Huber[1,2], Konrad U Förstner[3,4], Kai Papenfort[1,2,5]***

[1]Friedrich Schiller University Jena, Institute of Microbiology, Jena, Germany; [2]Faculty of Biology I, Ludwig-Maximilians-University of Munich, Martinsried, Germany; [3]TH Köln - University of Applied Sciences, Institute of Information Science, Cologne, Germany; [4]ZB MED - Information Centre for Life Sciences, Cologne, Germany; [5]Microverse Cluster, Friedrich Schiller University Jena, Jena, Germany

**Abstract** Negative feedback regulation, that is the ability of a gene to repress its own synthesis, is the most abundant regulatory motif known to biology. Frequently reported for transcriptional regulators, negative feedback control relies on binding of a transcription factor to its own promoter. Here, we report a novel mechanism for gene autoregulation in bacteria relying on small regulatory RNA (sRNA) and the major endoribonuclease, RNase E. TIER-seq analysis (transiently-inactivating-an-endoribonuclease-followed-by-RNA-seq) revealed ~25,000 RNase E-dependent cleavage sites in *Vibrio cholerae*, several of which resulted in the accumulation of stable sRNAs. Focusing on two examples, OppZ and CarZ, we discovered that these sRNAs are processed from the 3' untranslated region (3' UTR) of the *oppABCDF* and *carAB* operons, respectively, and base-pair with their own transcripts to inhibit translation. For OppZ, this process also triggers Rho-dependent transcription termination. Our data show that sRNAs from 3' UTRs serve as autoregulatory elements allowing negative feedback control at the post-transcriptional level.

**\*For correspondence:**
kai.papenfort@uni-jena.de

**Competing interests:** The authors declare that no competing interests exist.

## Introduction

Biological systems function on a mechanism of inputs and outputs, each triggered by and triggering a specific response. Feedback control (a.k.a. autoregulation) is a regulatory principle wherein the output of a system amplifies (positive feedback) or reduces (negative feedback) its own production. Negative feedback regulation is ubiquitous among biological systems and belongs to the most thoroughly characterized network motifs (*Nitzan et al., 2017*; *Shen-Orr et al., 2002*). At the gene regulatory level, negative feedback control has been qualitatively and quantitatively studied. Most commonly, a transcription factor acts to repress its own transcription by blocking access of RNA polymerase to the promoter region. This canonical mode of negative autoregulation is universally present in living systems and in *Escherichia coli* more than 40% of the known transcription factors are controlled by this type of regulation (*Rosenfeld et al., 2002*). Several characteristics have been attributed to negative autoregulatory circuits including an altered response time and improved robustness towards fluctuations in transcript production rates (*Alon, 2007*).

More recently, the mechanisms underlying RNA-based gene regulation have also been investigated for their regulatory principles and network functions (*Nitzan et al., 2017*; *Pu et al., 2019*). In bacteria, small regulatory RNAs (sRNAs) constitute the largest class of RNA regulators and frequently bind to one of the major RNA-binding proteins, Hfq or ProQ. Hfq- and ProQ-associated sRNAs usually act by base-pairing with *trans*-encoded target mRNAs affecting translation initiation and transcript stability (*Holmqvist and Vogel, 2018*; *Kavita et al., 2018*). The sRNAs frequently target multiple transcripts and given that regulation can involve target repression or activation, it has

become ever more clear that sRNAs can rival transcription factors with respect to their regulatory scope and function (*Hör et al., 2018*).

Another key factor involved in post-transcriptional gene regulation is ribonuclease E (RNase E), an essential enzyme in *E. coli* and related bacteria required for ribosome biogenesis and tRNA maturation (*Mackie, 2013*). RNase E's role in sRNA-mediated expression control is manifold and includes the processing of sRNAs into functional regulators (*Chao et al., 2017*; *Dar and Sorek, 2018a*; *Papenfort et al., 2015a*; *Updegrove et al., 2019*; *Chao et al., 2012*) as well as the degradation of target transcripts (*Massé et al., 2003*; *Morita et al., 2005*). Inhibition of RNase E-mediated cleavage through sRNAs can stabilize the target transcript and activate gene expression (*Fröhlich et al., 2013*; *Papenfort et al., 2013*; *Richards and Belasco, 2019*).

Global transcriptome analyses have revealed the presence of numerous sRNAs produced from 3' UTRs (untranslated regions) of mRNAs, a significant subset of which requires RNase E for their maturation (*Adams and Storz, 2020*). These 3' UTR-derived sRNAs can be produced from monocistronic (*Chao and Vogel, 2016*; *Grabowicz et al., 2016*; *Huber et al., 2020*; *Wang et al., 2020*) as well as long, operonic mRNAs (*Davis and Waldor, 2007*; *De Mets et al., 2019*; *Miyakoshi et al., 2019*) and typically act to regulate multiple target mRNAs in trans. The RNase E C-terminus also provides the scaffold for a large protein complex, called the degradosome, which in the major human pathogen, *Vibrio cholerae*, has recently been implicated in the turn-over of hypomodified tRNA species (*Kimura and Waldor, 2019*).

The present work addresses the regulatory role of RNase E in *V. cholerae* at a genome-wide level. To this end, we generated a temperature-sensitive variant of RNase E in *V. cholerae* and employed TIER-seq (transiently-inactivating-an-endoribonuclease-followed-by-RNA-seq) to globally map RNase E cleavage sites (*Chao et al., 2017*). Our analyses identified ~25,000 RNase E-sensitive sites and revealed the presence of numerous stable sRNAs originating from the 3' UTR of coding sequences. Detailed analyses of two of these sRNAs, OppZ and CarZ, showed that 3' UTR-derived sRNAs can act in an autoregulatory manner to reduce the expression of mRNAs produced from the same genetic locus. The molecular mechanism of sRNA-mediated gene autoregulation likely involves inhibition of translation initiation by the sRNA followed by Rho-dependent transcription termination. This setup directly links the regulatory activity of the sRNAs to their de novo synthesis, analogous to their transcription factor counterparts. However, we show that, in contrast to transcriptional regulators, autoregulatory RNAs can act at a subcistronic level to allow discoordinate operon expression.

## Results

### TIER-seq analysis of *V. cholerae*

The catalytic activity of RNase E (encoded by the *rne* gene) is critical for many bacteria, including *V. cholerae* (*Cameron et al., 2008*). To study the role of RNase E in this pathogen, we mutated the DNA sequence of the *V. cholerae* chromosome encoding leucine 68 of RNase E to phenylalanine (*Figure 1—figure supplement 1*). This mutation is analogous to the originally described N3071 *rne*TS isolate of *E. coli* (*Apirion and Lassar, 1978*) and exhibits full RNase E activity at permissive temperatures (30°C), but is rendered inactive under non-permissive temperatures (44°C). We validated our approach by monitoring the expression of two known substrates of RNase E in *V. cholerae*: A) 5S rRNA, which is processed by RNase E from the 9S precursor rRNA (*Papenfort et al., 2015b*), and B) the MicX sRNA, which contains two RNase E cleavage sites (*Davis and Waldor, 2007*). For both RNAs, transfer of the wild-type strain to 44°C only mildly effected their expression, whereas the equivalent procedure performed with the *rne*TS strain led to the accumulation of the 9S precursor and the full-length MicX transcript (*Figure 1A*, lanes 1–2 vs. 3–4). Additionally, accumulation of the two RNase E-dependent processing intermediates of MicX was reduced in the *rne*TS strain at the non-permissive temperature.

These results showed that we successfully generated a temperature-sensitive RNase E variant in *V. cholerae* and enabled us to employ TIER-seq to determine RNase E-dependent cleavage sites at a global scale. To this end, we cultivated *V. cholerae* wild-type and *rne*TS strains at 30°C to late exponential phase (OD$_{600}$ of 1.0), divided the cultures in half and continued incubation for 60 min at either 30°C or 44°C. Total RNA was isolated and subjected to deep sequencing. We obtained ~187 million reads from the twelve samples (corresponding to three biological replicates of each strain

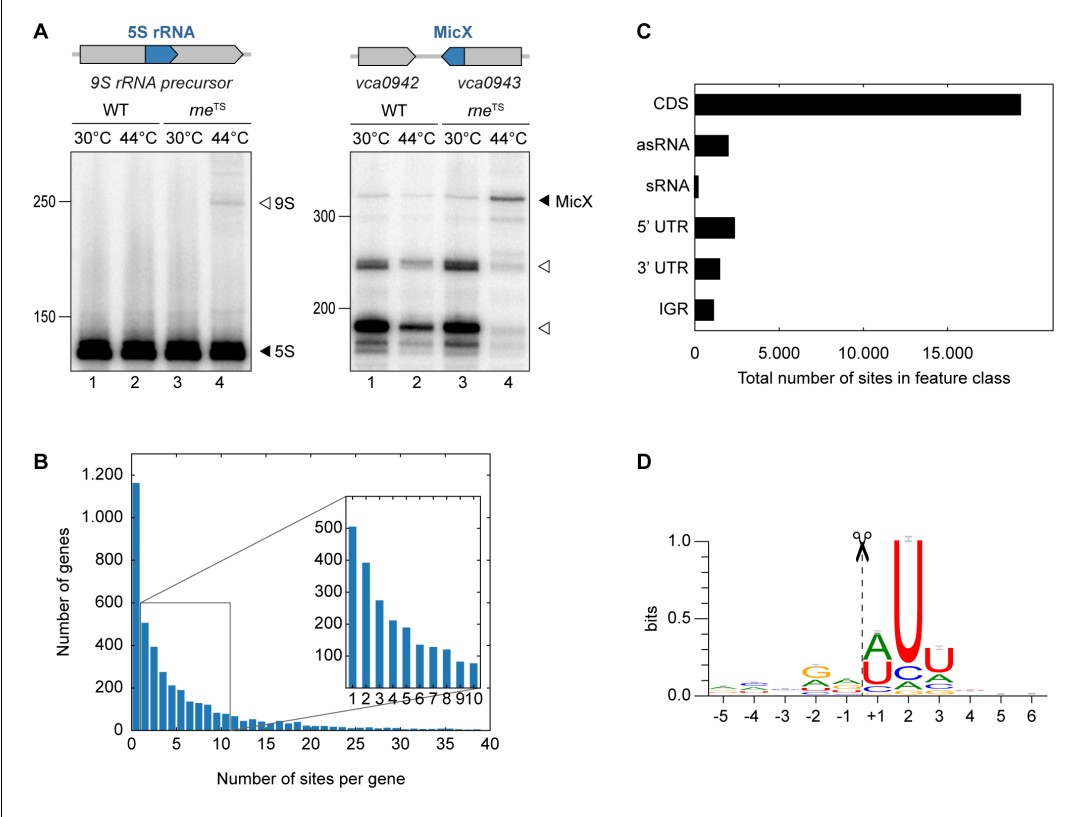

**Figure 1.** TIER-seq analysis of *V. cholerae*. (**A**) *V. cholerae* wild-type and *rne*TS strains were grown at 30°C to stationary phase (OD$_{600}$ of 2.0). Cultures were divided in half and continuously grown at either 30°C or 44°C for 60 min. Cleavage patterns of 5S rRNA and 3′ UTR-derived MicX were analyzed on Northern blots. Closed triangles indicate mature 5S or full-length MicX, open triangles indicate the 9S precursor or MicX processing products. (**B, C, D**) Biological triplicates of *V. cholerae* wild-type and *rne*TS strains were grown at 30°C to late exponential phase (OD$_{600}$ of 1.0). Cultures were divided in half and continuously grown at either 30°C or 44°C for 60 min. Isolated RNA was subjected to RNA-seq and RNase E cleavage sites were determined as described in the materials and methods section. (**B**) Number of cleavage sites detected per gene. (**C**) Classification of RNase E sites by their genomic location. (**D**) The RNase E consensus motif based on all detected cleavage sites. The total height of the error bar is twice the small sample correction. The online version of this article includes the following source data and figure supplement(s) for figure 1:

**Source data 1.** Full Northern blot images for the corresponding detail sections shown in *Figure 1* and RNase E cleavage site counts within genes or transcript categories.

**Figure supplement 1.** Conservation of RNase E between *E. coli* and *V. cholerae*.

**Figure supplement 2.** TIER-Seq read mapping statistics.

**Figure supplement 2—source data 1.** Number of obtained sequencing reads and Pearson correlation coefficients for library comparisons.

**Figure supplement 3.** Position and characteristics of RNase E cleavage sites.

**Figure supplement 4.** RNase E-mediated maturation of sRNAs from 3′ UTRs.

**Figure supplement 4—source data 1.** Full Northern blot images for the corresponding detail sections shown in *Figure 1—figure supplement 4*.

**Figure supplement 5.** RNase E-mediated maturation of sRNAs from IGRs.

**Figure supplement 5—source data 1.** Full Northern blot images for the corresponding detail sections shown in *Figure 1—figure supplement 5*.

**Figure supplement 6.** Expression of RNase E-independent sRNAs.

**Figure supplement 6—source data 1.** Full Northern blot images for the corresponding detail sections shown in *Figure 1—figure supplement 6*.

and condition; *Figure 1—figure supplement 2A*), resulting in ~98 million unique 5′ ends mapping to the *V. cholerae* genome. Comparison of the 5′ ends detected in wild-type and *rne*TS at 30°C showed almost no difference between the two strains (Pearson correlation coefficients R$^2$ ranging from 0.82 to 0.99 depending on the compared replicates), whereas the same analysis at 44°C revealed 24,962 depleted sites in the *rne*TS strain (*Figure 1—figure supplement 2B–C*). Given that γ-proteobacteria such as *V. cholerae* do not encode 5′ to 3′ exoribonucleases (*Mohanty and Kushner, 2018*), we designated these positions as RNase E-specific cleavage sites (*Supplementary file 1*).

Next, we analysed the ~25,000 RNase E sites with respect to frequency per gene and their distribution among different classes of transcript. We discovered that RNase E cleavage sites occur with a frequency of 2.8 (median)/6.3 (mean) sites per kb (*Figure 1B*). The majority of cleavage events occurs in coding sequences (~69.1%), followed by 5' UTRs (~8.4%), antisense RNAs (~7.1%), 3' UTRs (~5.3%), intergenic regions (~4.0%), and sRNAs (~0.6%) (*Figure 1C*). RNase E sites were slightly enriched around start and stop codons of mRNAs (*Figure 1—figure supplement 3A*). Furthermore, cleavage coincided with an increase in AU-content (*Figure 1—figure supplement 3B*) and a rise in minimal folding energies (*Figure 1—figure supplement 3C*), suggesting reduced secondary structure around RNase E sites. Together, these data allowed us to determine a consensus motif for RNase E in *V. cholerae* (*Figure 1D*). This 5-nt sequence, *i.e.* 'RN↓WUU', is highly similar to previously determined RNase E motifs of *Salmonella enterica* (*Chao et al., 2017*) and *Rhodobacter sphaeroides* (*Förstner et al., 2018*), indicating that RNase E operates by a conserved mechanism of recognition and cleavage.

## RNase E-mediated maturation of sRNAs

Earlier work on sRNA biogenesis in bacteria revealed that the 3' UTR of coding transcripts can serve as source for non-coding regulators and that RNase E is frequently required to cleave the sRNA from the mRNA (*Miyakoshi et al., 2015*). In *V. cholerae*, we previously annotated 44 candidate sRNAs located in the 3' UTR of mRNAs (*Papenfort et al., 2015b*). To analyse which of these sRNAs depend on RNase E for maturation, we searched for RNase E-cleavage sites matching with the first three bases of the annotated sRNAs. 17 sRNAs revealed potential RNase E-dependent maturation (*Supplementary file 2A*) and using Northern blot analyses of wild-type and *rne*[TS] samples, we were able to confirm these results for 9 sRNAs (Vcr016, Vcr041, Vcr044, Vcr045, Vcr053, Vcr064, FarS, Vcr079, and Vcr084; *Figure 1—figure supplement 4*). In all cases, transfer of the *rne*[TS] strain to non-permissive temperatures led to a change in mature sRNA levels and/or their upstream processing intermediates. We also discovered several sRNAs undergoing maturation by RNase E (*Supplementary file 2B*). Specifically, Northern blot analysis of Vcr043, Vcr065, and Vcr082 revealed that these sRNAs accumulate as multiple stable intermediates (*Figure 1—figure supplement 5*) that may contain different regulatory capacities as previously described for ArcZ and RprA of *S. enterica* (*Chao et al., 2017*; *Papenfort et al., 2015a*; *Soper et al., 2010*). In addition, we also analysed the expression of several RNase E-independent sRNAs (RyhB, Spot 42 and VqmR; *Figure 1—figure supplement 6*) on Northern blots. Inactivation of RNase E did not affect the levels of the mature sRNAs or any processed intermediates.

## OppZ is produced from the *oppABCDF* 3' end

To understand the regulatory functions of 3' UTR-derived sRNAs in *V. cholerae*, we focussed on Vcr045, which is processed from the 3' end of the *oppABCDF* mRNA (encoding an oligopeptide transporter) and which we hence named OppZ. The *oppZ* gene is 52 bps long and conserved among the *Vibrios* (*Figure 2A*). RNase E-mediated cleavage of *oppABCDF* occurs immediately downstream of the *oppF* stop codon and using the *rne*[TS] strain, we were able to validate RNase E-dependent processing of OppZ (*Figure 2B*). Northern and Western blot analysis of a *V. cholerae* strain carrying a 3XFLAG epitope at the C-terminus of the chromosomal *oppA* and *oppB* genes revealed that OppZ expression coincided with the expression of both proteins (*Figure 2C*, lanes 1–4). Previous transcriptome data showed that expression of *oppABCDF* is controlled by a single promotor located ~120 bps upstream of *oppA* (*Papenfort et al., 2015b*), indicating that the sRNA is co-expressed with all five *opp* genes. To test this prediction, we replaced the native promoter upstream of the chromosomal *oppA* gene with the L-arabinose-inducible pBAD promoter and monitored OppA, OppB, and OppZ expression under inducing and non-inducing conditions. In the absence of the inducer, expression of OppA/B and OppZ was strongly reduced (*Figure 2C*, lanes 5–8) and L-arabinose had no effect on the activity of the native *oppA* promoter (*Figure 2C*, lanes 9–10). In contrast, activation of the pBAD promoter led to a significant increase in OppA/B and OppZ (*Figure 2C*, lanes 11–12), indicating that expression of the *oppABCDF-oppZ* operon is indeed controlled by a single promoter.

To support these results and confirm production of OppZ from the longer precursor transcript, we generated two plasmids carrying either only *oppZ* or *oppF-oppZ* under the control of the

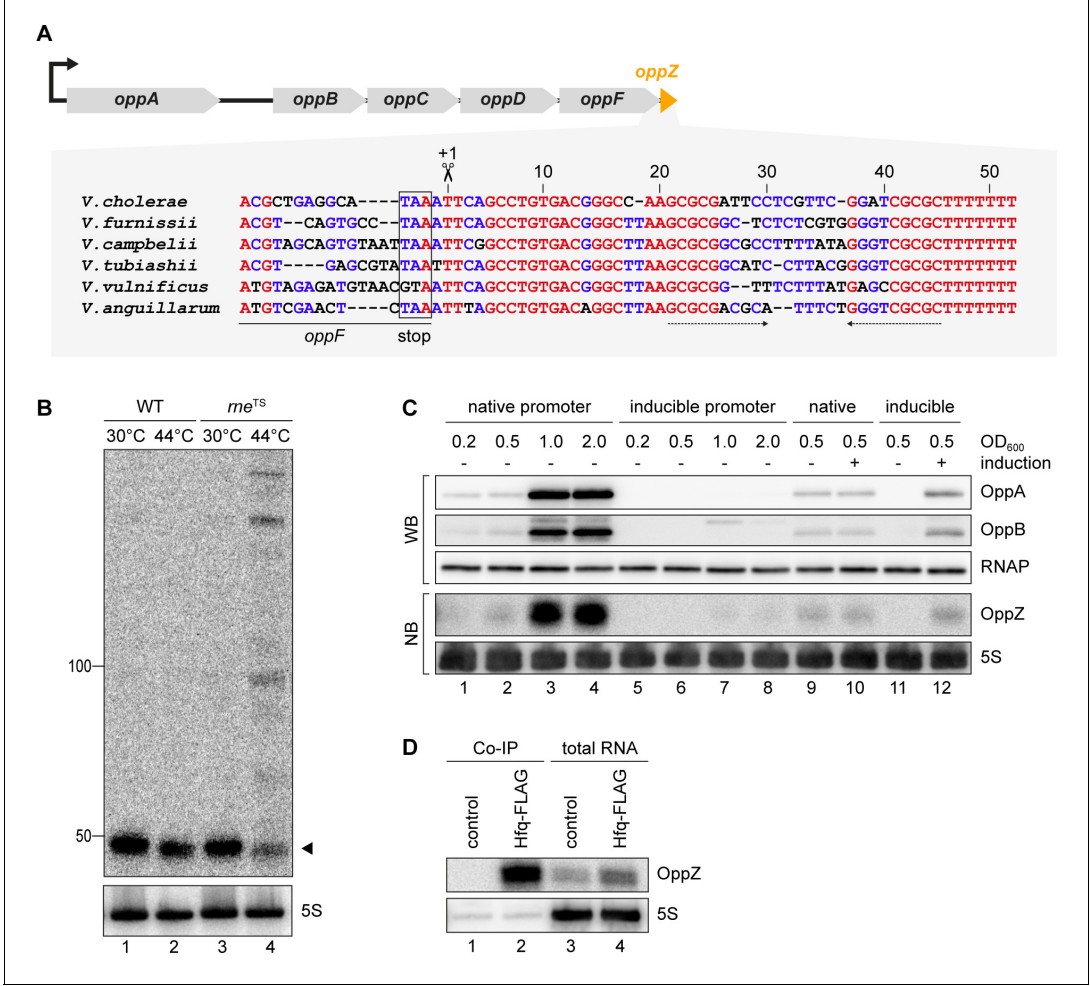

**Figure 2.** OppZ is produced from the *oppABCDF* 3' end. (A) Top: Genomic organization of *oppABCDF* and *oppZ*. Bottom: Alignment of *oppZ* sequences, including the last codons of *oppF*, from various *Vibrio* species. The *oppF* stop codon, the RNase E cleavage site and the Rho-independent terminator are indicated. (B) *V. cholerae* wild-type and *rne*TS strains were grown at 30°C to stationary phase (OD$_{600}$ of 2.0). Cultures were divided in half and continuously grown at either 30°C or 44°C for 30 min. OppZ synthesis was analyzed by Northern blot with 5S rRNA as loading control. The triangle indicates the size of mature OppZ. (C) Protein and RNA samples were obtained from *V. cholerae oppA*::3XFLAG *oppB*::3XFLAG strains carrying either the native *oppA* promoter or the inducible pBAD promoter upstream of *oppA*. Samples were collected at the indicated OD$_{600}$ and tested for OppA and OppB production by Western blot and for OppZ expression by Northern blot. RNAP and 5S rRNA served as loading controls for Western and Northern blots, respectively. Lanes 1–8: Growth without L-arabinose. Lanes 9–12: Growth with either H$_2$O (-) or L-arabinose (+) (0.2% final conc.). (D) *V. cholerae* wild-type (control) and *hfq*::3XFLAG (Hfq-FLAG) strains were grown to stationary phase (OD$_{600}$ of 2.0), lysed, and subjected to immunoprecipitation using the anti-FLAG antibody. RNA samples of lysate (total RNA) and co-immunoprecipitated fractions were analyzed on Northern blots. 5S rRNA served as loading control.

The online version of this article includes the following source data and figure supplement(s) for figure 2:

**Source data 1.** Full Northern and Western blot images for the corresponding detail sections shown in *Figure 2*.
**Figure supplement 1.** Hfq dependence of OppZ processing.
**Figure supplement 1—source data 1.** Full Northern blot images for the corresponding detail sections shown in *Figure 2—figure supplement 1*.
**Figure supplement 2.** Hfq dependence of OppZ stability.
**Figure supplement 2—source data 1.** Quantification of OppZ levels in wild-type and Δhfq cells from Northern blots.

constitutive P$_{Tac}$ promoter (*Figure 2—figure supplement 1A*) and compared OppZ expression in wild-type and Δ*oppZ* cells. Expression of mature OppZ was readily detected from the precursor (*Figure 2—figure supplement 1B*, lane 1 vs. 4) and the size of the processed OppZ transcript was comparable to endogenously expressed OppZ (lane 1) and OppZ transcribed directly by the P$_{Tac}$ promoter (lane 3). We also repeated these experiments in a *V. cholerae hfq* mutant (*Svenningsen et al., 2009*). Here, processing of the precursor into OppZ was still detected

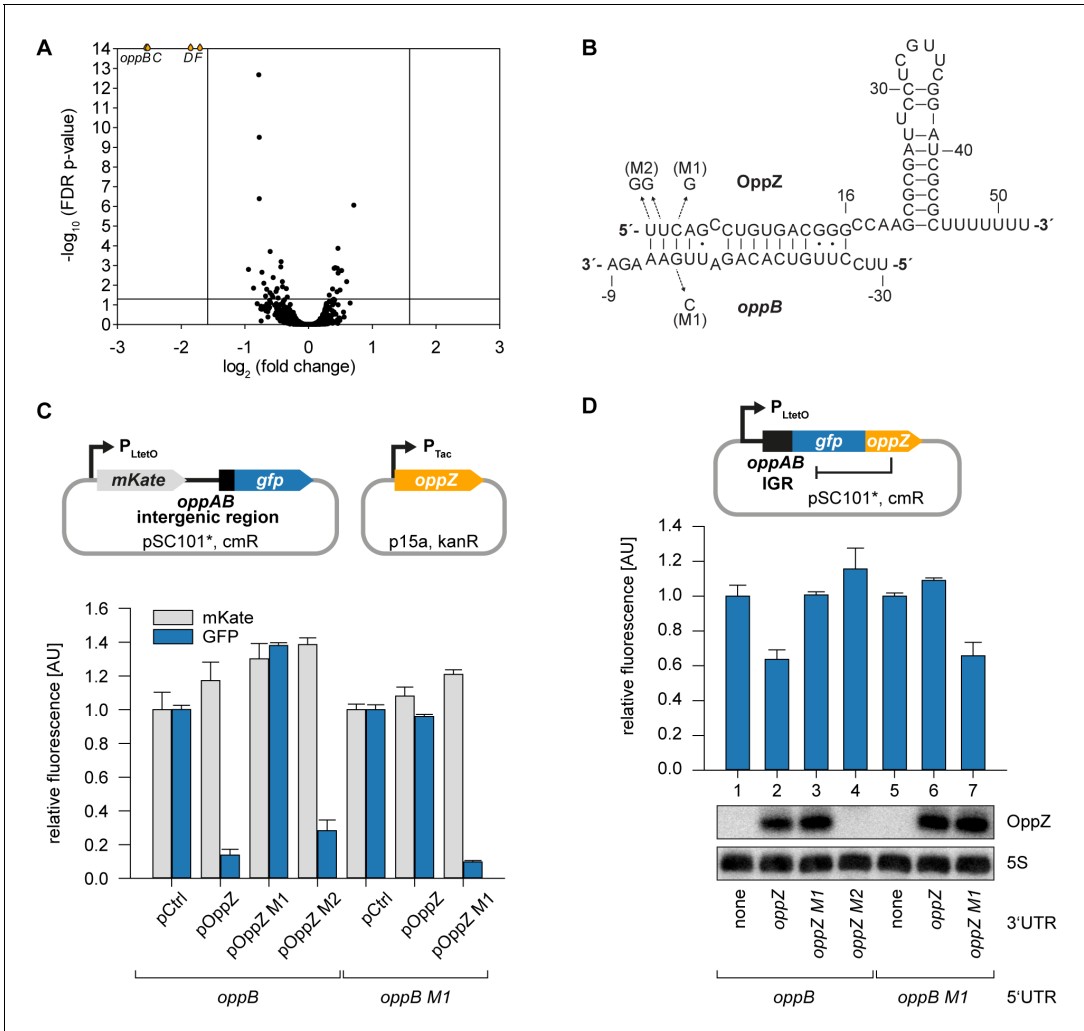

**Figure 3.** Feedback autoregulation at the suboperonic level. (A) Volcano plot of genome-wide transcript changes in response to inducible OppZ over-expression. Lines indicate cut-offs for differentially regulated genes at 3-fold regulation and FDR-adjusted p-value≤0.05. Genes with an FDR-adjusted p-value<$10^{-14}$ are indicated as droplets at the top border of the graph. (B) Predicted OppZ secondary structure and base-pairing to *oppB*. Arrows indicate the mutations tested in (C) and (D). (C) *E. coli* strains carrying a translational reporter plasmid with the *oppAB* intergenic region placed between *mKate2* and *gfp* were co-transformed with a control plasmid or the indicated OppZ expression plasmids. Transcription of the reporter and *oppZ* were driven by constitutive promoters. Cells were grown to $OD_{600}$ = 1.0 and fluorophore production was measured. mKate and GFP levels of strains carrying the control plasmid were set to 1. Error bars represent the SD of three biological replicates. (D) Single-plasmid regulation was measured by inserting the indicated *oppZ* variant into the 3' UTR of a translational *oppB::gfp* fusion. Expression was driven from a constitutive promoter. *E. coli* strains carrying the respective plasmids were grown to $OD_{600}$ = 1.0 and GFP production was measured. Fluorophore levels from control fusions without an sRNA gene were set to one and error bars represent the SD of three biological replicates. OppZ expression was tested by Northern blot; 5S rRNA served as loading control.

The online version of this article includes the following source data and figure supplement(s) for figure 3:

**Source data 1.** Full Northern blot images for the corresponding detail sections shown in *Figure 3* and raw data for fluorescence measurements.

**Figure supplement 1.** Pulse expression of OppZ reduces *oppBCDF* transcript levels.

**Figure supplement 1—source data 1.** Full Northern blot images for the corresponding detail sections shown in *Figure 3—figure supplement 1* and raw data for transcript changes as determined by qRT-PCR.

**Figure supplement 2.** Hfq-dependent, post-transcriptional repression of OppBCDF by OppZ.

**Figure supplement 2—source data 1.** Full Northern blot images for the corresponding detail sections shown in *Figure 3—figure supplement 2* and raw data for fluorescence measurements.

**Figure supplement 3.** Mutational analysis of the RNase E site in *oppZ*.

**Figure supplement 3—source data 1.** Full Northern blot images for the corresponding detail sections shown in *Figure 3—figure supplement 3*.

(lane 8), however, the steady-state levels of OppZ were lower, suggesting that OppZ binds Hfq. Indeed, stability experiments using rifampicin-treated *V. cholerae* showed that OppZ half-life is reduced in Δ*hfq* cells (*Figure 2—figure supplement 2*), and RNA co-immunoprecipitation experiments of chromosomal Hfq::3XFLAG revealed that OppZ interacts with Hfq in vivo (*Figure 2D*). Together, these data show that OppZ is an Hfq-dependent sRNA that is processed from the 3' UTR of the polycistronic *oppABCDF* mRNA by RNase E.

## Feedback Autoregulation at the suboperonic level

Hfq-binding sRNAs control gene expression by base-pairing with *trans*-encoded target transcripts (*Kavita et al., 2018*). To determine the targets of OppZ in *V. cholerae*, we cloned the sRNA (starting from the RNase E cleavage site) on a plasmid under the control of the pBAD promoter. Induction of the pBAD promoter for 15 min resulted in a strong increase in OppZ levels (~30 fold, *Figure 3—figure supplement 1A*) and RNA-seq experiments of the corresponding samples revealed four repressed genes (*Figure 3A* and *Figure 3—figure supplement 1B*). Interestingly, these genes were *oppBCDF*, i.e. the same transcript that OppZ is processed from. We validated OppZ-mediated repression of all four genes using qRT-PCR (*Figure 3—figure supplement 1C*), which also confirmed that the first gene of the operon, *oppA*, is not affected by OppZ. Despite the reduced transcript levels of *oppBCDF*, OppZ over-expression did not reduce the stability of the *oppB* messenger (*Figure 3—figure supplement 1D*). Using the *RNA-hybrid* algorithm (*Rehmsmeier et al., 2004*), we were able to predict RNA duplex formation of the *oppB* translation initiation site with the 5' end of the OppZ sRNA (*Figure 3B*). We confirmed this interaction using a variant of a previously reported post-transcriptional reporter system (*Corcoran et al., 2012*). Here, the first gene of the operon is replaced by the red-fluorescent mKate2 protein, followed by the *oppAB* intergenic sequence and the first five codons of *oppB,* which were fused to *gfp* (*Figure 3C*, top). Transfer of this plasmid into *E. coli* and co-transformation of the OppZ over-expression plasmid resulted in strong repression of GFP (~7 fold), while mKate2 levels remained constant. Mutation of either OppZ or *oppB* (mutations M1, see *Figure 3B*) abrogated regulation of GFP and combination of both mutants restored control (*Figure 3C*, bottom). In contrast, OppZ-mediated repression of OppB::GFP was strongly reduced in *E. coli* lacking *hfq* (*Figure 3—figure supplement 2A–B*). We also generated three additional variants of the reporter plasmids in which we included the *oppBC*, *oppBCD*, and *oppBCDF* sequences fused to GFP (*Figure 3—figure supplement 2C*). In all cases, OppZ readily inhibited GFP but did not affect mKate2. These results confirm that OppZ promotes discoordinate expression of the *oppABCDF* operon.

Next, we aimed to reproduce OppZ-mediated repression from a single transcript. To this end, we compared GFP production of a translational *oppB::gfp* reporter with the same construct carrying the *oppZ* sequence downstream of *gfp* (*Figure 3D*, top). Northern blot analysis revealed that OppZ was efficiently clipped off from the *gfp* transcript in this construct and fluorescence measurements showed that OppZ also inhibited GFP expression (*Figure 3D*, bottom, lane 1 vs. 2). We confirmed that this effect is specific to base-pairing of OppZ with the *oppAB* intergenic sequence as we were able to recapitulate our previous compensatory base-pair exchange experiments using the single plasmid system (*Figure 3D*). In addition, mutation of the RNase E recognition site in *oppZ* (UU→GG, mutation M2; *Figure 3—figure supplement 3A*) blocked OppZ maturation and abolished OppB::GFP repression (*Figure 3D*, lane 4; *Figure 3—figure supplement 3B*), whereas expression of OppZ M2 from a separate plasmid efficiently reduced OppB:GFP levels (*Figure 3C*). Together, our data demonstrate that OppZ down-regulates protein synthesis from its own cistron. Furthermore, mutation M2 shows that this autoregulation is not mediated by long-distance intramolecular base-pairing of OppZ with the *oppB* 5' UTR, but rather requires RNase E-dependent maturation of the transcript followed by Hfq-dependent base-pairing.

## Translational control of OppZ synthesis

The above experiments revealed that OppZ inhibits protein production through feedback control, however, it was not clear if OppZ would also inhibit its own synthesis. To address this question, we generated an OppZ over-expression plasmid in which we mutated the sequence of the terminal stem-loop at eight positions. We call this construct '*regulator OppZ*' (*Figure 4A*). These mutations are not expected to inactivate the base-pairing function of OppZ, but will allow us to differentiate

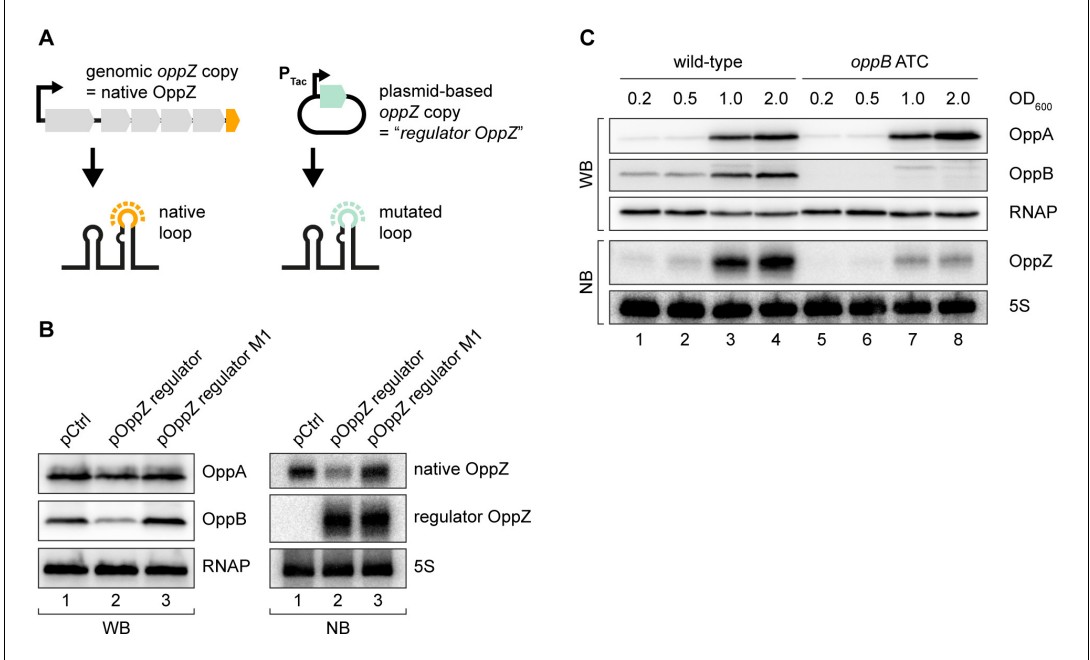

**Figure 4.** Translational control of OppZ synthesis.  (A) Schematic of the analyzed OppZ variants containing the native stem loop sequence (produced from the genomic *oppZ* locus) or a mutated stem loop sequence ('*regulator OppZ*' produced from a plasmid-based constitutive promoter). (B) *V. cholerae oppA*::3XFLAG *oppB*::3XFLAG carrying a control plasmid (pCMW-1) or a plasmid expressing *regulator OppZ* (pMD194, pMD195) were grown to stationary phase ($OD_{600}$ of 2.0). OppA and OppB production were tested by Western blot and expression of native OppZ and regulator OppZ was monitored on Northern blot using oligonucleotides binding to the respective loop sequence variants. RNAP and 5S rRNA served as loading controls for Western blot and Northern blot, respectively. (C) The *oppB* start codon was mutated to ATC in an *oppA*::3XFLAG *oppB*::3XFLAG background. *V. cholerae* strains with wild-type or mutated *oppB* start codon were grown in LB medium. Protein and RNA samples were collected at the indicated $OD_{600}$ and tested for OppA and OppB production by Western blot and for OppZ expression by Northern blot. RNAP and 5S rRNA served as loading controls for Western and Northern blots, respectively.

The online version of this article includes the following source data and figure supplement(s) for figure 4:

**Source data 1.** Full Northern and Western blot images for the corresponding detail sections shown in *Figure 4*.

**Figure supplement 1.** Translational control of OppZ synthesis.

**Figure supplement 1—source data 1.** Quantification of OppZ levels in wild-type and oppB ATC cells from Northern blots and full blot images for the corresponding detail sections shown in *Figure 4—figure supplement 1*.

the levels of native OppZ and *regulator OppZ* on Northern blots. Indeed, when tested in *V. cholerae*, over-expression of *regulator OppZ* inhibited OppB::3XFLAG production, but did not affect OppA::3XFLAG levels (*Figure 4B*, left). Importantly, *regulator OppZ* also reduced the expression of native OppZ (*Figure 4B*, right) and introduction of the M1 mutation (see *Figure 3B*) in *regulator OppZ* abrogated this effect. These results revealed that OppZ also exerts autoregulation of its own transcript.

Gene expression control by sRNAs typically occurs post-transcriptionally (*Gorski et al., 2017*) raising the question of how OppZ achieves autoregulation at the molecular level. Given that OppZ inhibits OppB production (*Figure 4B*), we hypothesized that OppZ synthesis might be linked to *oppB* translation. To test this prediction, we inactivated the chromosomal start codon of *oppB* (ATG→ATC) and monitored OppA/B and OppZ expression by Western and Northern blot, respectively. As expected, mutation of the *oppB* start codon had no effect on OppA::3XFLAG levels, but nullified OppB::3XFLAG production (*Figure 4C*, top). Lack of *oppB* translation also resulted in a strong decrease in OppZ levels (*Figure 4C*, bottom), however, did not change OppZ stability (*Figure 4—figure supplement 1A*). In addition, plasmid-based complementation of OppB::3XFLAG in the *oppB* start codon mutant failed to restore OppZ expression (*Figure 4—figure supplement 1B*), showing that OppZ production is independent of the cellular OppB levels. Based on these and the results above, we propose that autorepression of *oppBCDF-oppZ* must occur by a mechanism involving both translation inhibition, as well as transcription termination.

## OppZ promotes transcription termination through Rho

To explain the reduction of OppZ expression in the absence of *oppB* translation, we considered premature transcription termination as a possible factor. This hypothesis was supported by our finding that OppZ over-expression efficiently reduced *oppB* mRNA levels without significantly affecting transcript stability (*Figure 3—figure supplement 1C–D*). In *E. coli*, Rho protein accounts for a major fraction of all transcription termination events (*Ciampi, 2006*) and has previously been associated with the regulatory activity of Hfq-dependent sRNAs (*Bossi et al., 2012*; *Sedlyarova et al., 2016*; *Wang et al., 2015*). Rho is specifically inhibited by bicyclomycin (BCM; *Zwiefka et al., 1993*) and consequently we tested the effect of the antibiotic on OppZ expression in *V. cholerae* wild-type and the *oppB* start codon mutant. Whereas BCM had no effect on OppZ synthesis in wild-type cells (*Figure 5A*, lane 1 vs. 2), it strongly increased OppZ and *oppBCDF* expression in the absence of *oppB* translation (*Figure 5A*, lane 3 vs. 4, and *Figure 5B*). We confirmed these results by employing Term-Seq analysis (*Dar et al., 2016*) to wild-type and *oppB* start codon mutants cultivated with or without BCM. Detailed inspection of transcript coverage at the *oppABCDF-oppZ* genomic locus showed that lack of *oppB* translation down-regulated the expression of *oppBCDF-oppZ*, while presence of BCM suppressed this effect (*Figure 5C* and *Supplementary file 3B*). Similarly, inhibition of the *oppBCDF* mRNA and OppZ by over-expression of *regulator OppZ* (see *Figure 4A*) was suppressed in the presence of BCM, whereas OppB protein levels remained low presumably due to continued repression of *oppB* translation initiation by OppZ (*Figure 5D–E*).

To map the position of Rho-dependent transcription termination in *oppB*, we generated five additional strains carrying a STOP mutation at the 2nd, 15th, 65th, 115th, or 215th codon of the chromosomal *oppB* gene (*Figure 6A*). In addition, we mutated the start codons of *oppC*, *oppD*, and *oppF* and probed OppZ levels on Northern blot (*Figure 6B*). In accordance with the data presented in *Figure 4C*, mutation of the *oppB* start codon resulted in strongly decreased OppZ levels (*Figure 6B*, lane 1 vs. 2) and we observed similar results when the STOP mutation was introduced at the 2nd, 15th, and 65th codon of *oppB* (*Figure 6B*, lanes 3–5). In contrast, a STOP mutation at codon 115 led to increased OppZ expression (lane 6) and OppZ levels were fully restored when the STOP was placed at codon 215 of *oppB* (lane 7). Likewise, mutation of the *oppC*, *oppD*, and *oppF* start codons had no effect on OppZ production (*Figure 6B*, lanes 8–10). To summarize, our data indicate that autorepression of the *oppBCDF-oppZ* genes relies on inhibition of *oppB* translation initiation by OppZ, which triggers Rho-dependent transcription termination in the distal part of the *oppB* sequence.

## CarZ is another autoregulatory sRNA from *V. cholerae*

Our TIER-seq analysis revealed 17 3' UTR-derived sRNAs produced by RNase E-mediated cleavage in *V. cholerae* (*Supplementary file 2A*). Detailed analysis of OppZ showed that this sRNA serves as an autoregulatory element inhibiting the *oppBCDF* genes as well as its own synthesis (*Figures 4–6*). We therefore asked how wide-spread RNA-mediated autoregulation is and if the other 16 3' UTR-derived sRNAs might serve a similar function in *V. cholerae*. To this end, we searched for potential base-pairing sequences between the sRNAs and the translation initiation regions of their associated genes using the *RNA-hybrid* algorithm (*Rehmsmeier et al., 2004*). Indeed, we were able to predict stable RNA duplex formation between the Vcr084 sRNA (located in the 3' UTR of the *carAB* operon; encoding carbamoyl phosphate synthetase) and the 5' UTR of *carA*, which is the first gene of the operon (*Figure 7A–B*). In analogy to OppZ, we named this sRNA CarZ. Plasmid-borne expression of CarZ strongly inhibited GFP production from *carA::gfp* and *carAB::gfp* reporters in *E. coli* (*Figure 7—figure supplement 1A–B*) and we obtained similar results using a single transcript *carA::gfp::carZ* construct (*Figure 7C*). CarZ binds Hfq in vivo (*Figure 7—figure supplement 1C*) and repression of *carA::gfp* by CarZ requires Hfq, possibly due to reduced CarZ levels in the *hfq* mutant (*Figure 7—figure supplement 1D–E*). We validated the predicted interaction using compensatory base-pair exchange experiments (*Figure 7B–C*, *Figure 7—figure supplement 1A–B*). Transcription of *carAB-carZ* is controlled by a single promoter located upstream of *carA* and the three genes are co-expressed in vivo (*Figure 7D* and *Papenfort et al., 2015b*). These results suggested that CarZ provides feedback regulation and using an experimental strategy analogous to *Figure 4A*, we were able to show that CarZ inhibits CarA and CarB protein expression as well as its own synthesis (*Figure 7B,E*). Furthermore, introduction of a STOP codon at the 2nd codon of the chromosomal

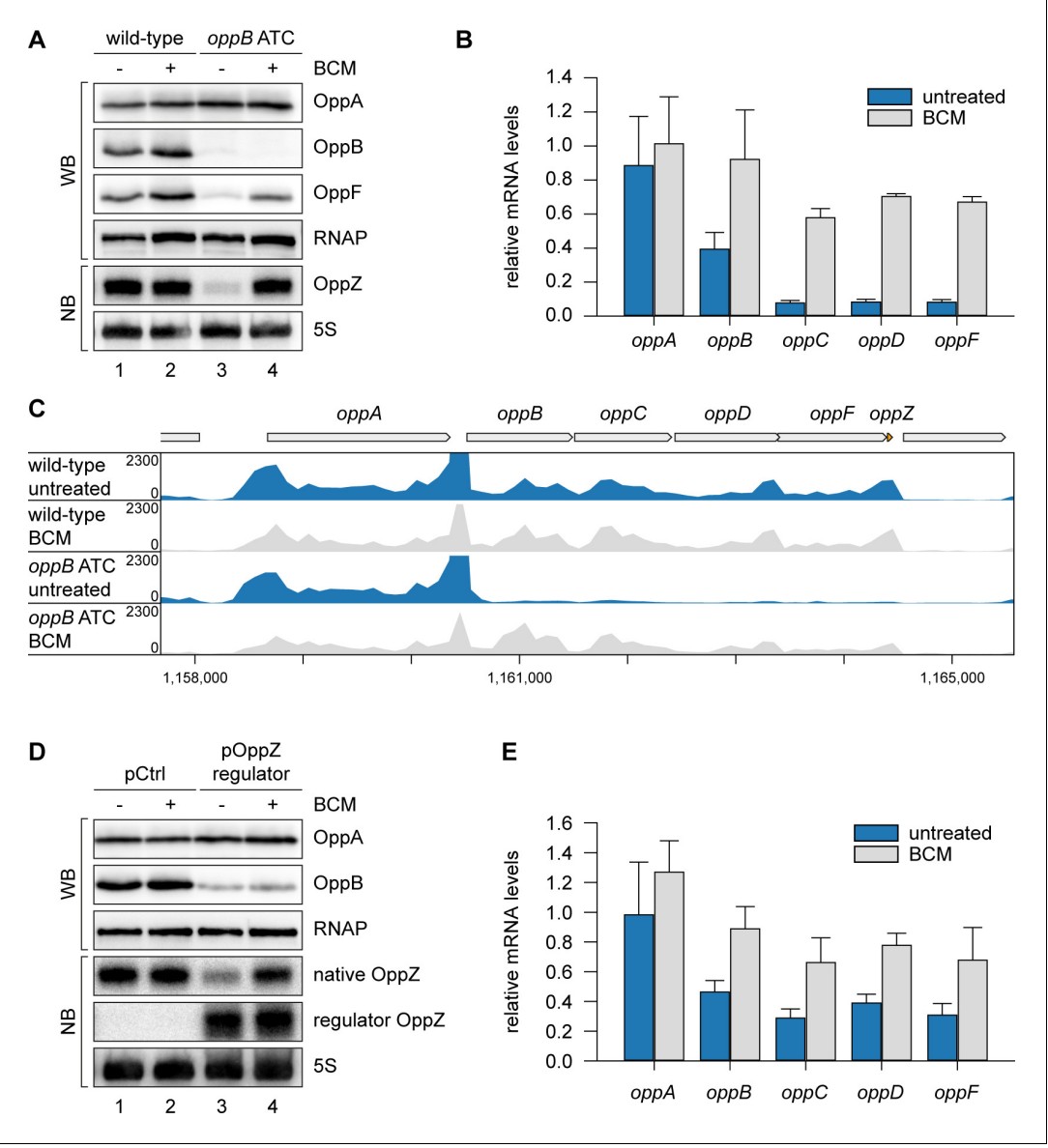

**Figure 5.** OppZ promotes transcription termination through Rho. (A) *V. cholerae oppA*::3XFLAG *oppB*::3XFLAG *oppF*::3XFLAG strains with wild-type or mutated *oppB* start codon were grown to early stationary phase (OD$_{600}$ of 1.5). Cultures were divided in half and treated with either H$_2$O or BCM (25 μg/ml final conc.) for 2 hr before protein and RNA samples were collected. OppA, OppB and OppF production were tested by Western blot and OppZ expression was monitored by Northern blot. RNAP and 5S rRNA served as loading controls for Western and Northern blots, respectively. (B) Biological triplicates of *V. cholerae oppA*::3XFLAG *oppB*::3XFLAG strains with wild-type or mutated *oppB* start codon were treated with BCM as described in (A). *oppABCDF* expression in the *oppB* start codon mutant compared to the wild-type control was analyzed by qRT-PCR. Error bars represent the SD of three biological replicates. (C) Triplicate samples from (B) were subjected to Term-seq and average coverage of the *opp* operon is shown for one representative replicate. The coverage cut-off was set at the maximum coverage of annotated genes. (D) *V. cholerae oppA*::3XFLAG *oppB*::3XFLAG strains carrying a control plasmid (pMD397) or a plasmid expressing *regulator OppZ* (pMD398) were treated with BCM as described in (A). OppA and OppB production were tested by Western blot and expression of native OppZ and regulator OppZ was monitored on Northern blot using oligonucleotides binding to the respective loop sequence variants. RNAP and 5S rRNA served as loading controls for Western and Northern blots, respectively. (E) Levels of *oppABCDF* in the experiment described in (D) were analyzed by qRT-PCR. Error bars represent the SD of three biological replicates. The online version of this article includes the following source data for figure 5:

**Source data 1.** Full blot images for the corresponding detail sections shown in *Figure 5* and raw data for transcript changes as determined by qRT-PCR.

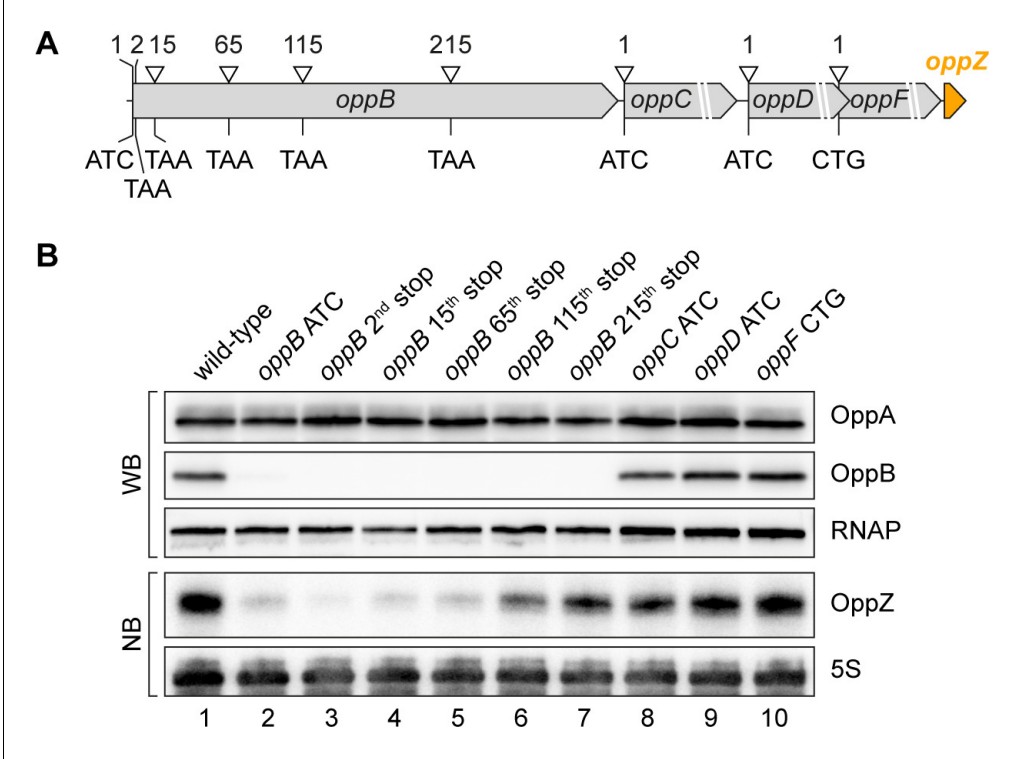

**Figure 6.** Influence of OppBCDF translation on OppZ expression. (**A**) The depicted mutations were individually inserted into the *opp* locus to inactivate the start codons of *oppB*, *oppC*, *oppD* or *oppF* or to insert STOP codons at the positions 2, 15, 65, 115 or 215 of *oppB*. (**B**) *V. cholerae oppA*::3XFLAG *oppB*::3XFLAG strains with the described *opp* mutations were grown: wild-type (lane 1), the *oppB* start codon mutated (lane 2), a STOP codon inserted at the 2nd, 15th, 65th, 115th or 215th codon of *oppB* (lanes 3–7) or mutated start codons of *oppC*, *oppD* or *oppF* (lanes 8–10). At stationary phase (OD$_{600}$ of 2.0), protein and RNA samples were collected and tested for OppA and OppB production by Western blot and for OppZ expression by Northern blot. RNAP and 5S rRNA served as loading controls for Western and Northern blots, respectively.

The online version of this article includes the following source data for figure 6:

**Source data 1.** Full Northern and Western blot images for the corresponding detail sections shown in *Figure 6*.

*carA* gene abrogated CarZ expression and similar results were obtained when the STOP codon was placed at the 2nd codon of *carB* (*Figure 7F*). Of note, inactivation of *carA* translation also blocked CarB production indicating, among other possibilities, that translation of the two ORFs might be coupled and that expression of CarZ relies on active translation of both ORFs. Together, these results provide evidence that CarZ is an autoregulatory sRNA and suggest that this function might be more wide-spread among the growing class of 3' UTR-derived sRNAs.

## Autoregulatory sRNAs modify the kinetics of gene induction

Bacterial sRNAs acting at the post-transcriptional level have recently been reported to add unique features to gene regulatory circuits, including the ability to promote discoordinate operon expression (*Nitzan et al., 2017*). Plasmid-borne over-expression of OppZ resulted in decreased expression of the *oppBCDF* cistrons, while leaving *oppA* levels unaffected (*Figure 3—figure supplement 1B–C*). We therefore asked if OppZ expression had a similar effect on the production of their corresponding proteins. To this end, we cultivated wild-type and *oppZ*-deficient *V. cholerae* (both carrying a control plasmid), as well as Δ*oppZ* cells carrying an OppZ over-expression plasmid, to various stages of growth and monitored OppA and OppB levels on Western Blot (*Figure 8—figure supplement 1A*). Quantification of the results revealed a moderate increase in OppB expression (~1.8 fold) in cells lacking *oppZ* and ~5 fold decreased OppB levels when OppZ was over-expressed. Neither lack of *oppZ*, nor OppZ over-expression significantly affected OppA production (*Figure 8—figure supplement 1B–C*).

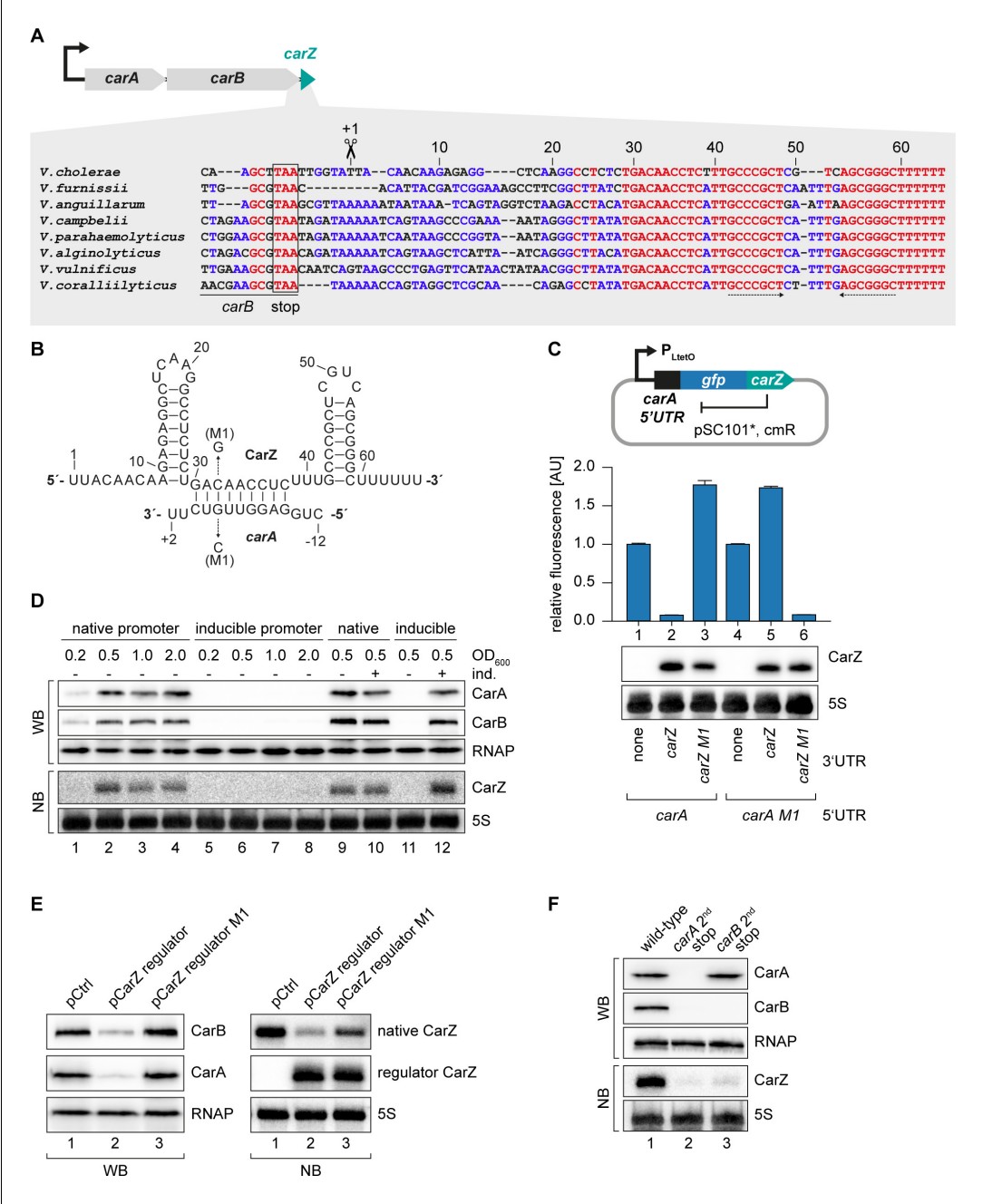

**Figure 7.** CarZ is another autoregulatory sRNA from *V. cholerae*. (**A**) Top: Genomic context of *carAB* and *carZ*. Bottom: Alignment of *carZ* sequences, including the last codons of *carB*, from various *Vibrio* species. The *carB* stop codon, the RNase E cleavage site and the Rho-independent terminator are indicated. (**B**) Predicted CarZ secondary structure and base-pairing to *carA*. Arrows indicate the single nucleotide mutations tested in (**C**). (**C**) Single-plasmid feedback regulation of *carA* by CarZ was measured by inserting the indicated *carZ* variant into the 3' UTR of a translational *carA::gfp* fusion. Expression was driven from a constitutive promoter. *E. coli* strains carrying the respective plasmids were grown to $OD_{600}$ = 1.0 and GFP production was measured. Fluorophore levels from control fusions without an sRNA gene were set to one and error bars represent the SD of three biological replicates. CarZ expression was tested by Northern blot; 5S rRNA served as loading control. (**D**) Protein and RNA samples were obtained from *V. cholerae carA*::3XFLAG *carB*::3XFLAG carrying either the native *carA* promoter or the inducible pBAD promoter upstream of *carA*. Samples were collected at the indicated $OD_{600}$ and tested for CarA and CarB production by Western blot and for CarZ expression by Northern blot. RNAP and 5S rRNA served as loading controls for Western and Northern blots, respectively. Lanes 1–8: Growth without L-arabinose. Lanes 9–12: Growth with either $H_2O$ (-) or L-arabinose (+) (0.2% final conc.). (**E**) *V. cholerae carA*::3XFLAG *carB*::3XFLAG strains carrying a control plasmid or a plasmid expressing a CarZ variant with a mutated stem loop (*regulator CarZ*) were grown to late exponential phase ($OD_{600}$ of 1.0). CarA and CarB production were tested by Western blot and expression of native CarZ or regulator CarZ was monitored on Northern blot using oligonucleotides binding to the respective loop sequence

*Figure 7 continued on next page*

Figure 7 continued

variants. RNAP and 5S rRNA served as loading controls for Western blot and Northern blot, respectively. (F) *V. cholerae carA*::3XFLAG *carB*::3XFLAG strains with the following *carA* or *carB* mutations were grown: wild-type (lane 1) or a STOP codon inserted at the 2nd codon of *carA* (lane 2) or *carB* (lane 3), respectively. At late exponential phase (OD$_{600}$ of 1.0), protein and RNA samples were collected and tested for CarA and CarB production by Western blot and for CarZ expression by Northern blot. RNAP and 5S rRNA served as loading controls for Western and Northern blots, respectively. The online version of this article includes the following source data and figure supplement(s) for figure 7:

**Source data 1.** Full blot images for the corresponding detail sections shown in *Figure 7* and raw data for fluorescence measurements.
**Figure supplement 1.** Hfq-dependent, post-transcriptional repression of CarA and CarB by CarZ.
**Figure supplement 1—source data 1.** Full Northern blot images for the corresponding detail sections shown in *Figure 4—figure supplement 1* and raw data for fluorescence measurements.
**Figure supplement 2.** CarZ induces *carAB* degradation.
**Figure supplement 2—source data 1.** Raw data for transcript changes as determined by qRT-PCR.

Given the relatively mild effect of *oppZ* deficiency on steady-state OppB protein levels (*Figure 8—figure supplement 1A*), we next investigated the role of OppZ on the dynamics of OppABCDF expression. Specifically, transcription factor-controlled negative autoregulation has been reported to affect the response time of regulatory networks (*Rosenfeld et al., 2002*) and we speculated that sRNA-mediated feedback control could have a similar effect. To test this hypothesis, we employed a *V. cholerae* strain in which we replaced the native promoter upstream of the chromosomal *oppA* gene with the L-arabinose-inducible pBAD promoter (see *Figure 2C*) and monitored the kinetics of OppA and OppB production in wild-type and Δ*oppZ* cells before and at several time-points post induction (*Figure 8A*). Whereas OppA protein accumulated equally in wild-type and *oppZ* mutants (*Figure 8B*), expression of OppB was significantly increased in Δ*oppZ* cells (*Figure 8C*). This effect was most prominent at later stages after induction (>30 min) and coincided with accumulation of OppZ (*Figure 8A*). Calculation of the OppB response time (50% of the maximal expression value) showed a significant delay in Δ*oppZ* cells (~78 min), when compared to the wild-type control (~52 min). We therefore conclude that alike transcription factors, autoregulatory sRNAs change the dynamics of their associated genes, however, in contrast to transcription factors, sRNAs act at the post-transcriptional level and can direct this effect towards a specific subgroup of genes within an operon.

## Discussion

Base-pairing sRNAs regulating the expression of *trans*-encoded mRNAs are a major pillar of gene expression control in bacteria (*Gorski et al., 2017*). Transcriptomic data obtained from various microorganisms have shown that sRNAs are produced from almost all genomic loci and that the 3' UTRs of coding genes are a hotspot for sRNAs acting through Hfq (*Adams and Storz, 2020*). Expression of 3' UTR-derived sRNAs can either occur by independent promoters, or by ribonucleolytic cleavage typically involving RNase E (*Miyakoshi et al., 2015*). In the latter case, production of the sRNA is intimately connected to the activity of the promoter driving the expression of the upstream mRNA, suggesting that the regulatory function of the sRNA is linked to the biological role of the associated genes. Indeed, such functional interdependence has now been demonstrated in several cases (*Chao and Vogel, 2016*; *De Mets et al., 2019*; *Huber et al., 2020*; *Miyakoshi et al., 2019*; *Wang et al., 2020*), however, it remained unclear if and how these sRNAs also affected their own transcripts. In this regard, OppZ and CarZ provide a paradigm for 3' UTR-derived sRNAs allowing autoregulation at the post-transcriptional level. This new type of feedback inhibition is independent of auxiliary transcription factors and we could show that autoregulation by sRNAs can either involve the full transcript (CarZ), or act at the suboperonic level (OppZ).

### Features of RNase E-mediated gene control

RNase E is a principal factor for RNA turnover in almost all Gram-negative bacteria (*Bandyra and Luisi, 2018*). The protein forms a tetramer in vivo and serves as the scaffold for the degradosome, a large, multi-enzyme complex typically containing the phosphorolytic exoribonuclease PNPase, the RNA-helicase RhlB, and the glycolytic enzyme enolase (*Aït-Bara and Carpousis, 2015*). Substrates of RNase E are preferentially AU-rich and harbor a 5' mono-phosphate. Thus, the enzyme relies on

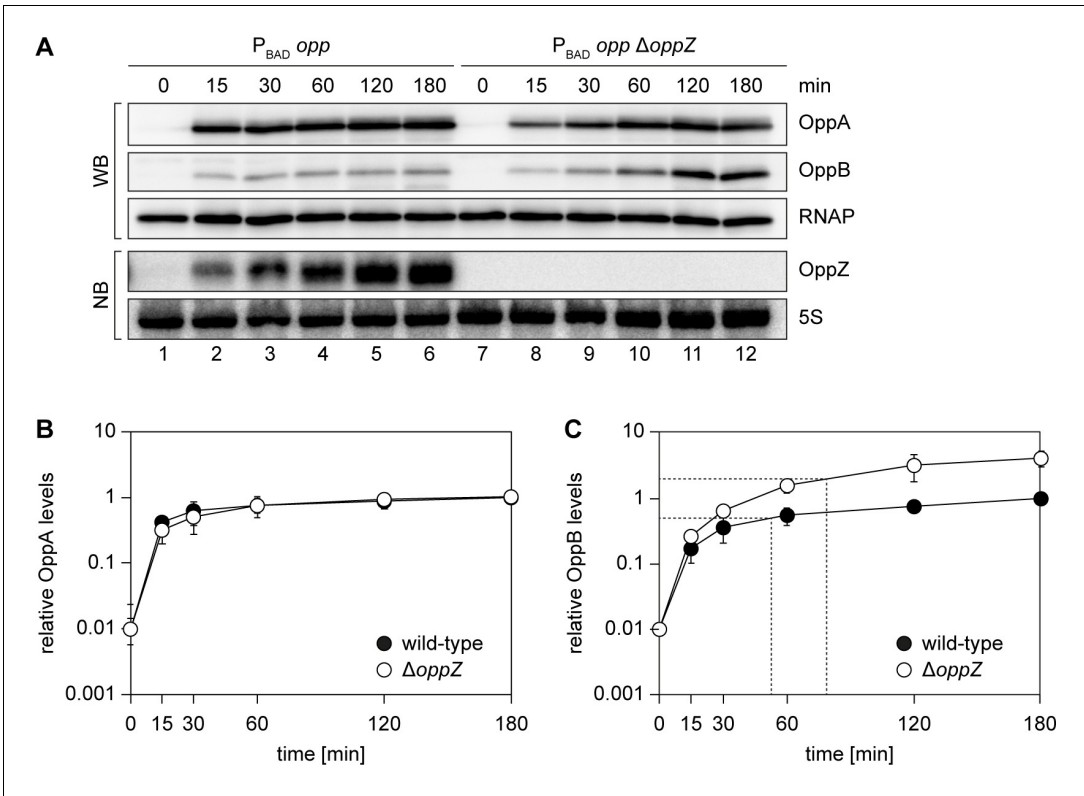

**Figure 8.** Modified kinetics of gene induction by autoregulatory OppZ. (**A**) Expression of the *opp* operon including the *oppA*::3XFLAG and *oppB*::3XFLAG genes and the native *oppZ* gene (lanes 1–6) or an *oppZ* deletion (lanes 7–12) was induced from the pBAD promoter at late exponential phase (OD$_{600}$ of 1.0) by the addition of L-arabinose (0.2% final conc.). Protein and RNA samples were obtained at the indicated time points and tested for OppA and OppB production by Western blot and for OppZ expression by Northern blot. RNAP and 5S rRNA served as loading controls for Western and Northern blots, respectively. (**B, C**) Quantification of OppA (**B**) or OppB (**C**) levels from the experiment in (**A**); error bars represent the SD of three biological replicates. Data are presented as fold regulation of OppA or OppB in Δ*oppZ* compared to the wild-type. Dashed lines in (**C**) indicate the time points of half-maximum OppB expression.

The online version of this article includes the following source data and figure supplement(s) for figure 8:

**Source data 1.** Quantification of OppAB protein levels from Western blots and full blot images for the corresponding detail sections shown in *Figure 8*.

**Figure supplement 1.** OppZ-dependent repression of OppA and OppB protein levels.

**Figure supplement 1—source data 1.** Quantification of OppAB protein levels from Western blots and full blot images for the corresponding detail sections shown in *Figure 8—figure supplement 1*.

RNA pyrophosphohydrolases such as RppH, which convert the 5' terminus from a triphosphate to a monophosphate, before transcript degradation can be initiated (*Deana et al., 2008*). Recognition of a substrate is followed by scanning of RNase E for suitable cleavage sites along the transcript (*Richards and Belasco, 2019*). TIER-seq-based identification of a consensus sequence for RNase E target recognition revealed highly similar motifs for *V. cholerae* (*Figure 1D*) and *S. enterica* (*Chao et al., 2017*). These results further support the previously proposed 'U$_{+2}$ Ruler-and-Cut' mechanism, in which a conserved uridine located two nts down-stream of the cleavage site is key for RNase E activity. However, in contrast to the data obtained from *S. enterica*, we discovered only a mild enrichment of RNase E cleavage sites occurring at translational stop codons (*Figure 1—figure supplement 3A*). This observation might be explained by differences in stop codon usage between *V. cholerae* and *S. enterica* (*Korkmaz et al., 2014*) and could point to species-specific features of RNase E activity.

## The role of termination factor Rho in sRNA-mediated gene expression control

Approximately 25–30% of all genes in *E. coli* depend on Rho for transcription termination (*Cardinale et al., 2008*; *Dar and Sorek, 2018b*; *Peters et al., 2012*). BCM treatment of *V. cholerae* wild-type cells revealed 699 differentially regulated genes (549 upregulated and 150 repressed genes; *Supplementary file 3A*), suggesting an equally global role for Rho in this organism. Rho-dependent transcription termination is modulated by various additional factors (*Mitra et al., 2017*). This includes anti-termination factors such as NusG, as well as Hfq and its associated sRNAs (*Bossi et al., 2020*). For sRNAs, the effect on Rho activity can be either activating or repressing. Previous work has shown that sRNAs can mask Rho-dependent termination sites and thereby promote transcriptional read-through (*Lin et al., 2019*; *Sedlyarova et al., 2016*). Negative gene regulation involving sRNAs and Rho typically includes translation inhibition by the sRNA resulting in separation of transcription and translation complexes (*Figure 9*). Coupling of transcription and translation normally protects the nascent mRNA from Rho action and loss of ribosome binding supports

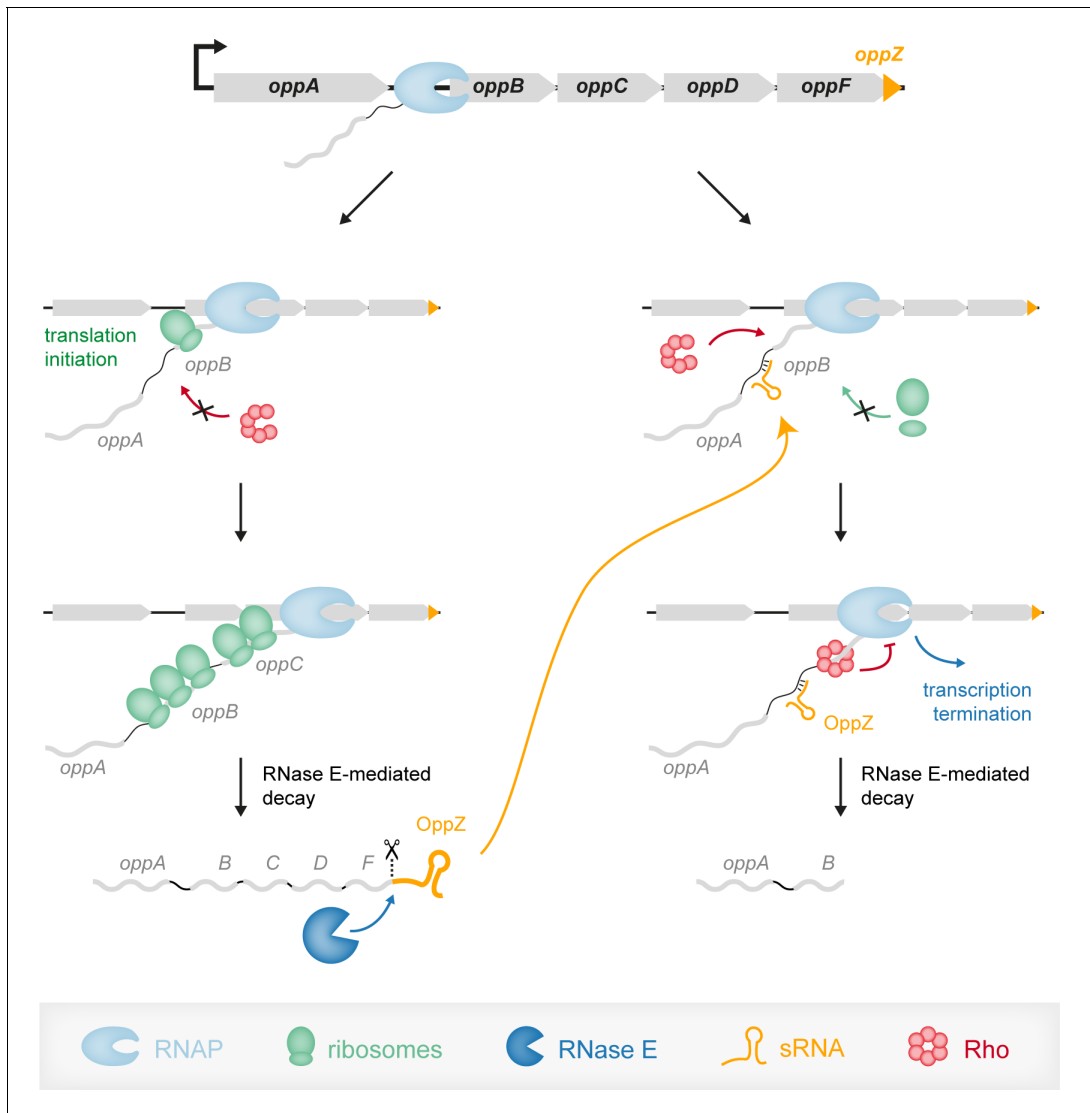

**Figure 9.** Model of the OppZ-dependent mechanism of *opp* regulation. Transcription of the *oppABCDF* operon initiates upstream of *oppA* and in the absence of OppZ (left) involves all genes of the operon as well as OppZ. In this scenario, all cistrons of the operon are translated. In the presence of OppZ (right), the sRNA blocks translation of *oppB* and the ribosome-free mRNA is recognized by termination factor Rho. Rho catches up with the transcribing RNAP and terminates transcription pre-maturely within *oppB*. Consequently, *oppBCDF* are not translated and OppZ is not produced.

transcription termination (*Bossi et al., 2012*). In addition, lack of ribosome-mediated protection can render the mRNA target vulnerable to ribonucleases, *e.g.* RNase E, which can also lead to the degradation of the sRNA (*Feng et al., 2015*; *Massé et al., 2003*). Which of these mechanisms are at play for a given sRNA-target mRNA pair is most often unknown and it is likely that both types of regulation can occur either independently or in concert. For example, over-expression of OppZ did not affect *oppB* transcript stability (*Figure 3—figure supplement 1D*), suggesting that induction of Rho-mediated transcription termination is the main mechanism for gene repression in this sRNA-target mRNA pair. In contrast, analogous experiments testing the stability of the *carA* and *carB* transcripts upon CarZ over-expression revealed a significant drop in transcript stability for both mRNAs (*Figure 7—figure supplement 2A–B*). These results suggest that translation inhibition of *carA* by CarZ has two outcomes: 1st) accelerated ribonucleolytic decay of the *carAB* transcript and 2nd) Rho-mediated transcription termination. Using two regulatory mechanisms (CarZ-*carA*) instead of one (OppZ-*oppB*) might explain the strong inhibition of *carA::gfp* by CarZ (~10 fold, *Figure 7C*), when compared to the relatively weak repression (1.8-fold) of *oppB::gfp* by OppZ (*Figure 3D*).

Employing multiple regulatory mechanisms on one target mRNA might have led to an underestimation of the prevalence of Rho-mediated transcription termination in sRNA-mediated gene control. In fact, sRNAs frequently repress genes that are downstream in an operon with their base-pairing target, which could point to a possible involvement of Rho (*Bossi et al., 2020*). Rho is known to bind cytosine-rich RNA elements (*Allfano et al., 1991*), however, due to the strong variability in size and composition of these sequences, predicting Rho binding sites (a.k.a. *rut* sites) from genomic or transcriptomic data has been a difficult task (*Nadiras et al., 2018*). Indeed, while our transcriptomic data of the *oppB* start codon mutant did not allow us to pinpoint the position of the *rut* site in *oppB* (*Figure 5C*), evidence obtained from genetic analyses using various *oppB* STOP codon mutants revealed that Rho-dependent termination likely occurs at or close to codon 115 in *oppB* (*Figure 6B*). We attribute the lack of this termination event in the transcriptomic data to the activity of 3′−5′ acting exoribonucleases (*e.g.* RNase II or PNPase *Bechhofer and Deutscher, 2019*; *Mohanty and Kushner, 2018*), which degrade the untranslated *oppB* sequence. Identifying the relevant exonucleases might well allow for an advanced annotation of global Rho-dependent termination sites and cross-comparison with documented sRNA-target interaction could help to clarify the relevance of Rho-mediated termination in sRNA-based gene control.

## Dynamics of RNA-based feedback regulation

Transcription factors and sRNAs are the principal components of gene networks. While the regulatory outcome of sRNA and transcription factor activity is often very similar, the underlying regulatory dynamics are not (*Hussein and Lim, 2012*). Regulatory networks involving sRNAs and transcription factors are called mixed circuits and have now been studied in greater detail. Similar to systems relying on transcription factors, feedback regulation is common among sRNAs (*Nitzan et al., 2017*). However, unlike the examples presented in this study, these circuits always involve the action of a transcription factor, which has implications for their regulatory dynamics. For example, the OmpR transcription factor activates the expression of the OmrA/B sRNAs, which repress their own synthesis by inhibiting the *ompR-envZ* mRNA (*Guillier and Gottesman, 2008*). This constitutes an autoregulatory loop, however, given that transcription of OmrA/B ultimately relies on OmpR protein levels, this regulation will only become effective when sufficient OmpR turn-over has been achieved (*Brosse et al., 2016*). In contrast, autoregulatory circuits involving 3′ UTR-derived sRNAs are independent of such auxiliary factors and therefore provide a more rapid response. In case of OppZ-*oppB*, we showed that the sRNA has a rapid effect on OppB expression levels (*Figure 8C*) and given the involvement of Rho-mediated transcription termination in this process, we expect similar dynamics for OppZ autoregulation (*Figure 9*).

Another key difference between feedback regulation by transcription factors and 3′ UTR-derived sRNAs is the stoichiometry of the players involved. In transcription factor-based feedback loops, the mRNA coding for the autoregulatory transcription factor can go through multiple rounds of translation, which will lead to an excess of the regulator over the target promoter. The degree of autoregulation is then determined by the cellular concentration of the transcription factor and the affinity towards its own promoter (*Rosenfeld et al., 2002*). In contrast, autoregulatory sRNAs which are generated by ribonucleolytic cleavage come at a 1:1 stoichiometry with their targets. However, this situation changes when the sRNA controls multiple targets. For OppZ, we have shown that

*oppBCDF* is the only transcript regulated by the sRNA (*Figure 3A*) and we currently do not know if CarZ has additional targets besides *carAB*. In addition, not all sRNA-target interactions result in changes in transcript levels as previously reported for the interaction of the Qrr sRNAs with the *luxO* transcript (*Feng et al., 2015*). New technologies, for example RIL-Seq (*Melamed et al., 2020*; *Melamed et al., 2016*), capturing the global interactome of base-pairing sRNAs independent of their regulatory state could help to address this question and clarify the stoichiometric requirements for sRNA-mediated autoregulation.

## Possible biological relevance of autoregulatory sRNAs

Autoregulation by 3' UTR-derived sRNAs allows for discoordinate operon expression, which is in contrast to their transcription factor counterparts. This feature might be particularly relevant for long mRNAs containing multiple cistrons, such as *oppABCDF*. The *oppABCDF* genes encode an ABC transporter allowing high affinity oligopeptide uptake (*Hiles et al., 1987*). OppBCDF constitute the membrane-bound, structural components of the transport system, whereas OppA functions as a periplasmic binding protein. The overall structure of the transporter requires each one unit of OppB, OppC, OppD, and OppF, while OppA does constitutively interact with the complex and typically accumulates to higher concentrations in the periplasm (*Doeven et al., 2004*). Given that transcription of *oppABCDF* is controlled exclusively upstream of *oppA* (*Figure 2C* and *Papenfort et al., 2015b*), OppZ-mediated autoregulation of *oppBCDF* (rather than the full operon) might help to achieve equimolar concentrations of OppB, OppC, OppD, and OppF in the cell without affecting OppA production.

The *carAB* genes, which are repressed by CarZ, encode carbamoyl phosphate synthetase; an enzyme complex catalyzing the first step in the separate biosynthetic pathways for the production of arginine, and pyrimidine nucleotides (*Castellana et al., 2014*). Similar to OppBCDF, the CarAB complex contains one subunit of CarA and one subunit of CarB. Transcriptional control of *carAB* is complex and controlled by several transcription factors integrating information from purine, pyrimidine, and arginine pathways (*Charlier et al., 2018*). While the exact biological role of CarZ-mediated feedback regulation of *carAB* requires further investigation, transcription factor-based feedback regulation has been reported to reduce transcriptional noise (*Alon, 2007*), which could also be an important feature of sRNA-mediated autoregulation. The OppZ and CarZ sRNAs identified in this study now provide the framework to test this prediction.

## Orthogonal use of gene autoregulation by 3' UTR-derived sRNAs

Regulatory RNAs have now been established as powerful components of the synthetic biology toolbox (*Qi and Arkin, 2014*). RNA regulators are modular, versatile, highly programmable, and therefore ideal candidates for synthetic biology approaches. Similarly, autoregulatory loops using transcriptional repressors find ample use in synthetic regulatory circuits (*Afroz and Beisel, 2013*). While it might be counterintuitive for a transcript to also produce its own repressor, negative feedback regulation has been reported to endow regulatory networks with improved robustness when disturbances to the system are imposed. Hfq-binding sRNAs providing feedback control have recently also been demonstrated to efficiently replace transcriptional regulation in artificial genetic circuits (*Kelly et al., 2018*). However, these sRNAs were produced from separate genes and therefore required additional transcriptional input, which increases noise. In contrast, the autoregulatory sRNAs presented here are produced by ribonucleolytic cleavage and we have shown that both OppZ and CarZ are efficiently clipped off from foreign genes, such as *gfp* (*Figure 3—figure supplement 3*, *Figure 7C*). We therefore propose that autoregulatory sRNAs can be attached to the 3' UTR of other genes as well, offering a simple and highly modular concept to introduce autoregulation into a biological system. These circuits can be further tuned by modifying the base-pairing strength of the RNA duplex formed between the sRNA and the target, as well as the introduction of Rho-dependent termination events. The latter could be used to avoid over-production of the sRNA, which will further shape the regulatory dynamics of the system. Given that transcriptomic analyses have revealed thousands of stable 3' UTR RNA tails derived from human transcripts (*Gruber and Zavolan, 2019*; *Malka et al., 2017*), we believe that RNA-based gene autoregulation also could be present and find applications in higher organisms.

# Materials and methods

**Key resources table**

| Reagent type (species) or resource | Designation | Source or reference | Identifiers | Additional information |
|---|---|---|---|---|
| Strain, strain background (*Escherichia coli*) | See *Supplementary file 4* | This study | | See *Supplementary file 4* |
| Strain, strain background (*Vibrio cholerae*) | See *Supplementary file 4* | This study | | See *Supplementary file 4* |
| Recombinant DNA reagent (plasmids) | See *Supplementary file 4* | This study | | See *Supplementary file 4* |
| Sequence-based reagent (oligonucleotides) | See *Supplementary file 4* | This study | | See *Supplementary file 4* |
| Antibody | ANTI-FLAG M2 antibody (mouse monoclonal) | Sigma-Aldrich | Cat#F1804; RRID:AB_262044 | (Western blot 1:1.000) |
| Antibody | RNA Polymerase alpha antibody 4RA2 (rabbit monoclonal) | BioLegend | Cat#WP003; RRID:AB_2687386 | (1:10.000) |
| Antibody | anti-mouse IgG HRP (goat polyclonal) | ThermoFischer | Cat#31430; RRID:AB_228307 | (1:10.000) |
| Antibody | anti-rabbit IgG HRP (goat polyclonal) | ThermoFischer | Cat#A16104; RRID:AB_2534776 | (1:10.000) |
| Commercial assay or kit | TURBO DNA-free Kit | Invitrogen | Cat#AM1907 | |
| Commercial assay or kit | NEBNext Ultra II Directional RNA Library Prep Kit for Illumina | NEB | Cat#E7760 | |
| Commercial assay or kit | Ribo-Zero rRNA Removal Kit (Gram-Negative Bacteria) | Illumina | Cat#MRZGN126 | |
| Chemical compound, drug | Protein G Sepharose | Sigma-Aldrich | Cat##P3296 | |
| Chemical compound, drug | Bicyclomycin (BCM) | SantaCruz Biotech. | Cat#sc-391755; CAS ID: 38129-37-2 | |
| Software, algorithm | MultAlin | *Corpet, 1988* (PMID:2849754) | | http://multalin.toulouse.inra.fr/multalin |
| Software, algorithm | RNAhybrid | *Rehmsmeier et al., 2004* (PMID:15383676) | | http://bibiserv2.cebitec.uni-bielefeld.de RRID:SCR_003252 |
| Software, algorithm | CLC Genomics Workbench | Qiagen | | https://qiagenbioinformatics.com RRID:SCR_011853 |
| Software, algorithm | SigmaPlot | SYSTAT | | https://systatsoftware.com RRID:SCR_003210 |

*Continued on next page*

*Continued*

| Reagent type (species) or resource | Designation | Source or reference | Identifiers | Additional information |
|---|---|---|---|---|
| Software, algorithm | GelQuantNET | biochemlabsolutions | | http://biochemlabsolutions.com/GelQuantNET.html RRID:SCR_015703 |
| Software, algorithm | BIO-1D | VILBER | | http://vilber.de/en/products/analysis-software |
| Software, algorithm | ImageJ | *Schneider et al., 2012* (PMID:22930834) | | https://imagej.nih.gov/ij/ RRID:SCR_003070 |
| Software, algorithm | cutadapt | *Martin, 2011* | | https://doi.org/10.14806/ej.17.1.200 |
| Software, algorithm | READemption | *Förstner et al., 2014* (PMID:25123900) | | https://doi.org/10.5281/zenodo.591469 |
| Software, algorithm | DESeq2 | *Love et al., 2014* (PMID:25516281) | | http://www.bioconductor.org/packages/release/bioc/html/DESeq2.html |
| Software, algorithm | RNAfold | *Lorenz et al., 2011* (PMID:22115189) | | http://www.tbi.univie.ac.at/RNA |
| Software, algorithm | WebLogo | *Crooks et al., 2004* (PMID:15173120) | | http://weblogo.threeplusone.com/ |
| Software, algorithm | BEDTools | *Quinlan and Hall, 2010* (PMID:20110278) | | http://code.google.com/p/bedtools |

## Strains, plasmids, and growth conditions

Bacterial strains, plasmids and DNA oligonucleotides used in this study are listed in *Supplementary file 4*. Throughout the study, *V. cholerae* C6706 (*Thelin and Taylor, 1996*) was used as the wild-type strain. *V. cholerae* and *E. coli* strains were grown aerobically in LB medium at 37°C except for temperature-sensitive strains. For stationary phase cultures of *V. cholerae*, samples were collected with respect to the time point when the cells reached an $OD_{600}$ >2.0, i.e., 3 hr after cells reached an $OD_{600}$ reading of 2.0. For transcript stability experiments, rifampicin was used at 250 µg/ml. To inhibit Rho-dependent transcription termination, bicyclomycin (BCM; sc-391755; Santa Cruz Biotechnology, Dallas, Texas) was used at 25 µg/ml. Other antibiotics were used at the following concentrations: 100 µg/ml ampicillin; 20 µg/ml chloramphenicol; 50 µg/ml kanamycin; 50 U/ml polymyxin B; and 5,000 µg/ml streptomycin.

For transient inactivation of RNase E, *V. cholerae* wild-type and a temperature-sensitive strain harboring the *rne-3071* mutation were grown at 30°C to the indicated cell density. Cultures were divided in half and either continuously grown at 30°C or shifted to 44°C. RNA samples were collected from both strains and temperatures at the indicated time points after the temperature shift.

RK2/RP4-based conjugal transfer was used to introduce plasmids into *V. cholerae* from *E. coli* S17λpir plasmid donor strains (*Simon et al., 1983*). Subsequently, transconjugants were selected using appropriate antibiotics and polymyxin B to specifically inhibit *E. coli* growth. *V. cholerae* mutant strains were generated as described previously (*Papenfort et al., 2015b*). Briefly, pKAS32 plasmids were transferred into *V. cholerae* strains by conjugation and cells were screened for ampicillin resistance. Single colonies were streaked on streptomycin plates for counter-selection and colonies were tested for desired mutations by PCR or sequencing. Strain KPEC53467 was generated by phage P1 transduction to transfer the Δ*hfq*::KanR allele (*Baba et al., 2006*) into *E. coli* Top 10 and subsequent removal of the KanR cassette using plasmid pCP20 *Datsenko and Wanner, 2000* following standard protocols.

## Plasmid construction

The plasmids used in this study are listed in *Supplementary file 4B*, used DNA oligonucleotides are listed in *Supplementary file 4C*. For pMD004, the *rrnB* terminator from pKP8-35 (*Papenfort et al., 2015b*) was amplified with KPO-1484/1485 and cloned by Gibson assembly into pKP-331 (*Papenfort et al., 2015b*) linearized with KPO-0196/1397. pMD089 was generated by amplification of *oppZ* from KPS-0014 chromosomal DNA using KPO-2552/2553 and Gibson assembly with pMD004 linearized with KPO-0196/1397. pMD373 was constructed by amplification of *oppB::3XFlag* from KPVC11709 chromosomal DNA using KPO-5878/5879 and Gibson assembly with pMD004 linearized with KPO-2789/pBAD-ATGrev. pCMW-2 was obtained by removing the promoterless *gfp* from pCMW-1 (*Waters and Bassler, 2006*) by amplification with KPO-2757/5421. pMD090 was generated by amplification of *oppZ* from KPS-0014 chromosomal DNA using KPO-2568/2553 and Gibson assembly with pEVS143 (*Dunn et al., 2006*) linearized with KPO-0092/1397. The M1 point mutation was introduced into pMD090 by site-directed mutagenesis with KPO-2619/2620, yielding pMD118. pMD194 and pMD195 were obtained by site-directed mutagenesis of pMD090 and pMD118, respectively, with KPO-3190/3191. pMD397 and pMD398 were obtained by replacing the p15a origin of replication in pCMW-1 and pMD194, respectively, by the pSC101 origin including an E93K mutation in the *repA* sequence. To this end, pCMW-1 and pMD194 were linearized with KPO-2041/2049, the pSC101 origin was amplified from pXG10-SF (*Corcoran et al., 2012*) in three parts (with KPO-6490/6493, KPO-6492/6495 and KPO-6494/6491) and fragments were joined with Gibson assembly. pMD173 and pMD174 were generated by amplification of the pBR322 origin from pBAD-Myc-His (Invitrogen) with KPO-2042/2043 and Gibson assembly with pCMW-1 or pMD090, respectively (both linearized with KPO-2041/2049). pMD197 was obtained by replacing the *oppZ* gene in pMD174 with a longer *oppF-oppZ* fragment (amplified from KPS-0014 chromosomal DNA using KPO-3197/2553) by Gibson assembly. pNP015 was constructed by amplification of *carZ* from KPS-0014 chromosomal DNA using KPO-1013/1014 and subcloning into linearized pEVS143 (KPO-0092/1023) with XbaI. Again, the M1 point mutation was introduced into pNP015 by site-directed mutagenesis with KPO-1782/1783, yielding pMH013. pMD361 and pMD362 were obtained by site-directed mutagenesis of pNP015 and pMH013, respectively, with KPO-5686/5687.

For translational GFP reporters, pMD093 was generated by amplification of the *oppAB* intergenic region and the first 5 codons of *oppB* from KPS-0014 chromosomal DNA using KPO-2580/2583 and Gibson assembly with pXG10-SF linearized with KPO-1702/1703. Site-directed mutagenesis of pMD093 with KPO-2615/2616 yielded pMD125. Accordingly, pMH010 and pMD374 were generated by amplification of the *carA* 5′UTR and the first 20 codons of *carA* with KPO-1674/1675 (for pMH010) or a fragment including the *carA* 5′ UTR, the complete *carA* gene and the first 20 codons of *carB* with KPO-1674/5874 (for pMD374) from KPS-0014 chromosomal DNA, followed by Gibson assembly with pXG10-SF linearized with KPO-1702/1703. Site-directed mutagenesis of pMH010 and pMD374 with KPO-1778/1779 yielded pMH012 and pMD375, respectively. For discoordinate translational reporters for *oppB* to *oppF*, fragments from the *oppAB* intergenic region to the first 5 codons of *oppB* or the first 20 codons of *oppC*, *oppD* or *oppF* were amplified from KPS-0014 chromosomal DNA using KPO-2622 and KPO-2583 (*oppB*), KPO-2577 (*oppC*), KPO-2578 (*oppD*) or KPO-2579 (*oppF*). *mKate2* was amplified from pMD079 (*Herzog et al., 2019*) with KPO-2511/2625 and the pXG10-SF backbone was linearized with KPO-2621/1703. Gibson assembly was used to join the pXG10-SF backbone, *mKate2* and the respective *opp* fragment to generate pMD120, pMD352, pMD353 and pMD354. Site-directed mutagenesis of pMD120 and pMD354 with KPO-2615/2616 yielded pMD129 and pMD355, respectively.

pMD091 and pMD112 were constructed by amplification of *oppZ* from KPS-0014 chromosomal DNA using KPO-2585/2586 and Gibson assembly with pXG10-SF (for pMD091) or pMD093 (for pMD112), both linearized with KPO-2584/2508. The M1 mutations in the *oppAB* IGR or *oppZ* were obtained by site-directed mutagenesis of pMD112 with KPO-2615/2616 or KPO-2617/2618, respectively, to construct pMD117, pMD127 and pMD128. Site-directed mutagenesis of pMD91 and pMD93 with KPO-2665/2666 to introduce the M2 mutation into *oppZ* yielded pMD124 and pMD126, respectively. Accordingly, pMD294 and pMD297 were constructed by amplification of *carZ* from KPS-0014 chromosomal DNA using KPO-4815/4817 and Gibson assembly with pMH010 (for pMD294) or pMH012 (for pMD297), both linearized with KPO-2584/2508. Site-directed mutagenesis of pMD294 and pMD297 with KPO-1782/1783 yielded pMD296 and pMD298, respectively.

All pKAS32-derived plasmids (*Skorupski and Taylor, 1996*) were constructed by Gibson assembly of the respective up and down flanks with the pKAS32 backbone (linearized with KPO-0267/0268) and an additional fragment containing the 3XFLAG sequence or an *araC*-pBAD fragment where appropriate. Flanks were amplified from KPS-0014 chromosomal DNA unless otherwise stated. Plasmids for gene deletions or chromosomal point mutations are listed in the following with the respective primer pairs for up and down flanks indicated: pMD003 (KPO-1440/1443 and KPO-1441/1442), pMD160 (KPO-2753/1199 and KPO-1200/2754), pMD350 (KPO-1429/1289 and KPO-1290/1430), pMD349 (KPO-5243/5244 from KPVC11709 chromosomal DNA and KPO-5245/5246), pMD357 (KPO-5243/5672 and KPO-5673/5246, both from KPVC11709 chromosomal DNA), pMD358 (KPO-5243/5674 and KPO-5675/5246, both from KPVC11709 chromosomal DNA), pMD370 (KPO-5880/5884 and KPO-5885/5881, both from KPVC11709 chromosomal DNA), pMD371 (KPO-5880/5886 and KPO-5887/5881, both from KPVC11709 chromosomal DNA), pMD372 (KPO-5882/5890 and KPO-5891/5883, both from KPVC11709 chromosomal DNA), pMD356 (KPO-3183/5670 and KPO-5671/3186, both from KPVC11709 chromosomal DNA), pMD367 (KPO-4395/5824 from KPVC11709 chromosomal DNA and KPO-5823/4400), pMD369 (KPO-4379/5828 and KPO-5827/4384), pMD385 (KPO-5235/6029 and KPO-6030/5238, both from KPVC12872 chromosomal DNA) and pMD386 (KPO-5223/6031 and KPO-6032/5226, both from KPVC12872 chromosomal DNA). For pMD199 and pMD200, flanks were amplified with KPO-3179/3180 and KPO-3181/3182 (for pMD199) or with KPO-3183/3184 and KPO-3185/3186 (for pMD200). The 3XFLAG fragment was obtained by annealing of the oligonucleotides KPO-3157/3158. Flanks and 3XFLAG tag for pMD269, pMD346 and pMD347 were amplified with the following oligonucleotides: KPO-4385/4386, KPO-4387/4388 and KPO-4389/4390 (for pMD269); KPO-5223/5224, KPO-5225/5226 and KPO-5231/5232 (for pMD346); KPO-5227/5228, KPO-5229/5230 and KPO-5233/5234 (for pMD347). pMD199 was used as template for the 3XFLAG fragments. For pMD280 and pMD351, a fragment containing the *araC* gene and the pBAD promoter was amplified from pMD004 using 4529/0196. Flanks were amplified with KPO-4527/4528 and KPO-4530/4531 (for pMD280) or with KPO-5235/5236 and KPO-5237/5238 (for pMD351).

## RNA isolation, Northern blot analysis and quantitative real-time PCR

For Northern blot analyses, total RNA was prepared and blotted as described previously (*Papenfort et al., 2017*). Membranes were hybridized in Roti-Hybri-Quick buffer (Carl Roth, Karlsruhe, Germany) with [$^{32}$P]-labeled DNA oligonucleotides at 42˚C or with riboprobes at 63˚C. Riboprobes were generated using the MAXIscript T7 Transcription Kit (Thermo Fisher Scientific, Waltham, Massachusetts). Signals were visualized using a Typhoon Phosphorimager (GE Healthcare, Chicago, Illinois) and quantified using GelQuant (RRID:SCR_015703; BioChemLabSolutions, San Francisco, California). Oligonucleotides for Northern blot analyses are provided in *Supplementary file 4C*. For qRT-PCR, total RNA was isolated with the SV Total RNA Isolation System (Promega, Fitchburg, Wisconsin). qRT–PCR was performed in three biological and two technical replicates using the Luna Universal One-Step RT-qPCR Kit (New England BioLabs, Ipswich, Massachusetts) and the MyiQ Single-Color Real-Time PCR Detection System (Bio-Rad, Hercules, California). 5S rRNA and *recA* were used as reference genes; oligonucleotides used for all qRT-PCR analyses are provided in *Supplementary file 4C*.

## Hfq co-immunoprecipitation

Hfq co-immunoprecipitations were performed as previously described (*Huber et al., 2020*). Briefly, *V. cholerae* wild-type (KPS-0014) and *hfq::3XFLAG* (KPS-0995) (*Peschek et al., 2019*) strains were grown in LB medium to OD$_{600}$ of 2.0. Lysates corresponding to 50 OD$_{600}$ units were subjected to immunoprecipitation using monoclonal anti-FLAG antibody (#F1804; Sigma-Aldrich, St. Louis, Missouri) and Protein G Sepharose (#P3296; Sigma-Aldrich).

## Western blot analysis and fluorescence assays

Total protein sample preparation and Western blot analyses were performed as described previously (*Papenfort et al., 2017*). Signals were visualized using a Fusion FX EDGE imager (Vilber Lourmat, Marne-la-Vallée, France) and band intensities were quantified using the BIO-1D software (Vilber Lourmat). 3XFLAG-tagged fusions were detected using mouse anti-FLAG antibody (#F1804; RRID:

AB_262044; Sigma-Aldrich) and goat anti-mouse HRP-conjugated IgG antibody, (#31430; RRID:AB_228307; Thermo Fisher Scientific). RNAPα served as a loading control and was detected using rabbit anti-RNAPα antibody (#WP003; RRID:AB_2687386; BioLegend, San Diego, California) and goat anti-rabbit HRP-conjugated IgG antibody, (#16104; AB_2534776; Thermo Fisher Scientific). Fluorescence assays of *E. coli* strains to measure mKate and GFP expression were performed as previously described (*Urban and Vogel, 2007*). Cells were washed in PBS and fluorescence intensity was quantified using a Spark 10 M plate reader (Tecan, Männedorf, Switzerland). Control strains not expressing fluorescent proteins were used to subtract background fluorescence.

## RNA-seq analysis: TIER-seq

*V. cholerae* wild-type and *rne*TS strains were grown in biological triplicates at 30°C to $OD_{600}$ of 1.0. Cultures were divided in half and either continuously grown at 30°C or shifted to 44°C. Cells were harvested from both strains and temperatures at 60 min after the temperature shift by addition of 0.2 volumes of stop mix (95% ethanol, 5% (v/v) phenol) and snap-frozen in liquid nitrogen. Total RNA was isolated and digested with TURBO DNase (Thermo Fisher Scientific). cDNA libraries were prepared by vertis Biotechnology AG (Freising, Germany): total RNA samples were poly(A)-tailed and 5'PPP structures were removed using RNA 5'Polyphosphatase (Epicentre, Madison, Wisconsin). An RNA adapter was ligated to the 5' monophosphate and first-strand cDNA synthesis was performed using an oligo(dT)-adapter and M-MLV reverse transcriptase. The resulting cDNAs were PCR-amplified, purified using the Agencourt AMPure XP kit (Beckman Coulter Genomics, Chaska, Minnesota) and sequenced using a NextSeq 500 system in single-read mode for 75 cycles.

After quality trimming and adapter clipping with cutadapt (version 2.5, DOI: https://doi.org/10.14806/ej.17.1.200) the sequencing reads were mapped to the *V. cholerae* reference genome (NCBI accession numbers: NC_002505.1 and NC_002506.1) including annotations for Vcr001-Vcr107 (*Papenfort et al., 2015b*) using READemption's (*Förstner et al., 2014*, v0.5.0, https://doi.org/10.5281/zenodo.591469) sub-command 'align' (building on segemehl version 0.3.4, *Hoffmann et al., 2009*) and nucleotide-specific coverage values were calculated with the sub-command 'coverage' based on the first base of the reads. Positions with a coverage of 20 reads or more were used to perform an enrichment analysis using DESeq2 (v.1.20.0, *Love et al., 2014*) comparing the WT to the mutant libraries. Nucleotides for which DESeq2 calculated an absolute fold-change of 3.0 or more and an adjusted (Benjamini-Hochberg corrected) p-value of 0.05 were treated in following analysis steps as bona fide cleavage sites.

The Minimum free energy (MFE) of sequence windows was computed with RNAfold (version 2.4.14) of the Vienna package (*Lorenz et al., 2011*). Sequence logos were created with WebLogo (version 3.7.4; *Crooks et al., 2004*). Overlaps of cleavage sites with other features were found by BEDTools' (version 2.26.0, *Quinlan and Hall, 2010*) sub-command 'intersect'. Pair-wise Pearson correlation coefficients between all samples were calculated based on the above mentioned first-base-in read coverages taking positions with a total sum of at least 10 reads in all samples combined into account. Positions that represent outliers with coverage values above the 99.99 percentile in one or more read libraries were not considered. The values were computed using the function 'corr' of the pandas Dataframe class (https://doi.org/10.5281/zenodo.3509134). For further details, please see the analysis scripts linked in the data and code availability section.

## RNA-seq analysis: Identification of OppZ targets

*V. cholerae* strains carrying either pBAD1K-ctrl or pBAD1K-*oppZ* were grown in biological triplicates to $OD_{600}$ of 0.5 and treated with 0.2% L-arabinose (final conc.). Cells were harvested after 15 min by addition of 0.2 volumes of stop mix (95% ethanol, 5% (v/v) phenol) and snap-frozen in liquid nitrogen. Total RNA was isolated and digested with TURBO DNase (Thermo Fisher Scientific). Ribosomal RNA was depleted using the Ribo-Zero kit for Gram-negative bacteria (#MRZGN126; Illumina, San Diego, California) and RNA integrity was confirmed with an Agilent 2100 Bioanalyzer. Directional cDNA libraries were prepared using the NEBNext Ultra II Directional RNA Library Prep Kit for Illumina (#E7760; NEB). The libraries were sequenced using a HiSeq 1500 System in single-read mode for 100 cycles. The read files in FASTQ format were imported into CLC Genomics Workbench v11 (RRID:SCR_011853; Qiagen, Hilden, Germany) and trimmed for quality and 3' adaptors. Reads were mapped to the *V. cholerae* reference genome (NCBI accession numbers: NC_002505.1 and

NC_002506.1) including annotations for Vcr001-Vcr107 (*Papenfort et al., 2015b*) using the 'RNA-Seq Analysis' tool with standard parameters. Reads mapping in CDS were counted, and genes with a total count cut-off >15 in all samples were considered for analysis. Read counts were normalized (CPM), and transformed (log2). Differential expression was tested using the built-in tool corresponding to edgeR in exact mode with tagwise dispersions ('Empirical Analysis of DGE'). Genes with a fold change ≥3.0 and an FDR-adjusted p-value≤0.05 were considered as differentially expressed.

### RNA-seq analysis: Bicyclomycin-dependent transcriptomes

*V. cholerae oppA*::3XFLAG *oppB*::3XFLAG *oppF*::3XFLAG strains with wild-type or mutated *oppB* start codon were grown in biological triplicates to $OD_{600}$ of 1.5, divided in half and treated with either bicyclomycin (25 µg/ml final conc.) or water. Cells were harvested after 120 min by addition of 0.2 volumes of stop mix (95% ethanol, 5% (v/v) phenol) and snap-frozen in liquid nitrogen. Total RNA was isolated and digested with TURBO DNase (Thermo Fisher Scientific). cDNA libraries were prepared by vertis Biotechnology AG in a 3' end-specific protocol: ribosomal RNA was depleted and the Illumina 5' sequencing adaptor was ligated to the 3' OH end of RNA molecules. First strand synthesis using M-MLV reverse transcriptase was followed by fragmentation and strand-specific ligation of the Illumina 3' sequencing adaptor to the 3' end of first-strand cDNA. Finally, 3' cDNA fragments were amplified, purified using the Agencourt AMPure XP kit (Beckman Coulter Genomics) and sequenced using a NextSeq 500 system in single-read mode for 75 cycles. The read files in FASTQ format were imported into CLC Genomics Workbench v11 (Qiagen) and trimmed for quality and 3' adaptors. Reads were mapped to the *V. cholerae* reference genome (NCBI accession numbers: NC_002505.1 and NC_002506.1) including annotations for Vcr001-Vcr107 (*Papenfort et al., 2015b*) using the 'RNA-Seq Analysis' tool with standard parameters. Reads mapping in CDS were counted, and genes with a total count cut-off >8 in all samples were considered for analysis. Read counts were normalized (CPM), and transformed (log2). Differential expression was tested using the built in tool corresponding to edgeR in exact mode with tagwise dispersions ('Empirical Analysis of DGE'). Genes with a fold change ≥3.0 and an FDR-adjusted p-value≤0.05 were considered as differentially expressed.

TIER-seq input data, analysis scripts and results are deposited at Zenodo (https://doi.org/10.5281/zenodo.3750832). Further information and requests for resources and reagents should be directed to and will be fulfilled by the corresponding author, Kai Papenfort (kai.papenfort@uni-jena.de).

## Acknowledgements

We thank Helmut Blum for help with the RNA sequencing experiments and Andreas Starick for excellent technical support. We thank Jörg Vogel, Gisela Storz, and Kathrin Fröhlich for comments on the manuscript and all members of the Papenfort lab for insightful discussions and suggestions.

## Additional information

### Funding

| Funder | Grant reference number | Author |
|---|---|---|
| Deutsche Forschungsgemeinschaft | GRK2062/1 | Mona Hoyos Kai Papenfort |
| Vallee Foundation | Vallee Scholars program | Kai Papenfort |
| H2020 European Research Council | StG-758212 | Michaela Huber Kai Papenfort |
| Human Frontier Science Program | CDA00024/2016-C | Mona Hoyos Michaela Huber Kai Papenfort |
| Deutsche Forschungsgemeinschaft | EXC 2051 | Mona Hoyos Kai Papenfort |
| Deutsche Forschungsgemeinschaft | 390713860 | Mona Hoyos Kai Papenfort |

The funders had no role in study design, data collection and interpretation, or the decision to submit the work for publication.

### Author contributions
Mona Hoyos, Conceptualization, Data curation, Formal analysis, Validation, Investigation, Visualization, Methodology, Writing - review and editing; Michaela Huber, Data curation, Validation, Investigation, Methodology, Writing - review and editing; Konrad U Förstner, Resources, Data curation, Software, Formal analysis, Writing - review and editing; Kai Papenfort, Conceptualization, Data curation, Supervision, Funding acquisition, Investigation, Methodology, Writing - original draft, Project administration, Writing - review and editing

### Author ORCIDs
Mona Hoyos  https://orcid.org/0000-0003-1085-4723
Kai Papenfort  https://orcid.org/0000-0002-5560-9804

### Decision letter and Author response
Decision letter https://doi.org/10.7554/eLife.58836.sa1
Author response https://doi.org/10.7554/eLife.58836.sa2

# Additional files

### Supplementary files
• Supplementary file 1. TIER-seq sites in *Vibrio cholerae*.
• Supplementary file 2. RNase E-mediated maturation of sRNAs.
• Supplementary file 3. BCM-sensitive transcripts in *Vibrio cholerae*.
• Supplementary file 4. Bacterial strains, plasmids and DNA oligonucleotides.
• Transparent reporting form

### Data availability
All high-throughput sequencing data was deposited at GEO: GSE148675 (TIER-seq), GSE144479 (OppZ target identification) and GSE144478 (Term-Seq analysis). We have uploaded source data for all figures and figure supplements showing the numerical data from our TIER-seq analysis, the raw data from GFP and mKate fluorescence measurements, fold changes obtained from qRT-PCR experiments and fold changes obtained by the quantification of Western and Northern blots. Additionally, for all figures showing cropped images of Western or Northern blots, we show the full image and indicate the cropped area and the antibody or labelled oligonucleotide used to detect the signal.

The following datasets were generated:

| Author(s) | Year | Dataset title | Dataset URL | Database and Identifier |
|---|---|---|---|---|
| Hoyos M, Huber M, Förstner K, Papenfort K | 2020 | Identification of bicyclomycin-sensitive transcripts in Vibrio cholerae | https://www.ncbi.nlm.nih.gov/geo/query/acc.cgi?acc=GSE144478 | NCBI Gene Expression Omnibus, GSE144478 |
| Hoyos M, Huber M, Förstner KU, Papenfort K | 2020 | Global identification of RNase E sites in Vibrio cholerae | https://www.ncbi.nlm.nih.gov/geo/query/acc.cgi?acc=GSE148675 | NCBI Gene Expression Omnibus, GSE148675 |
| Hoyos M, Huber M, Förstner K, Papenfort K | 2020 | OppZ target identification | https://www.ncbi.nlm.nih.gov/geo/query/acc.cgi?acc=GSE144479 | NCBI Gene Expression Omnibus, GSE144479 |

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
