## [Decision Letter]

**Acceptance summary:**

Bacteria often express regulatory RNAs that function by base-pairing with target mRNAs, leading to altered translation and/or RNA stability. Here, Hoyos et al. identify a novel feedback mechanism, whereby bacterial regulatory RNAs derived from mRNA 3' untranslated regions can negatively regulate their own expression.

**Decision letter after peer review:**

Thank you for submitting your article "Gene autoregulation at the RNA level" for consideration by *eLife*. Your article has been reviewed by three peer reviewers, including Joe Wade as the Reviewing Editor and Reviewer #1, and the evaluation has been overseen by Kevin Struhl as the Senior Editor.

The reviewers have discussed the reviews with one another, and the Reviewing Editor has drafted this decision to help you prepare a revised submission.

All the reviewers found the work to be of high quality, and were excited by the discovery of autoregulation of 3' UTR-derived sRNAs. The one aspect where we felt the data fell short of supporting the conclusions relates to the mechanism of autoregulation. Specifically, we felt that more experimental evidence is needed to support the idea that OppZ repression of oppB translation leads to Rho termination within oppB. The data clearly demonstrate Rho termination when the oppB start codon is mutated, but there are insufficient data to conclude that repression of oppB by OppZ has the same effect, especially in light of published data suggesting that translational repression needs to be strong to permit Rho termination (PMID: 19059415). As detailed below, it should be straightforward to experimentally test this model more rigorously. You should also respond to the other reviewer's comments, none of which require additional experiments.

Essential revision requiring new experimental data:

Provide additional experimental support for the model in which OppZ represses translation of oppB, leading to premature Rho termination. Specifically, test whether OppZ represses its own expression when BCM is added. As an example, you could repeat the experiment shown in Figure 4B but with BCM added. It would be useful to look at RNA levels of several opp genes, including oppB and oppZ, and OppB protein levels.

Essential revisions not requiring new experimental data:

1) Show the RNA-seq data from Figure 3 for the opp operon. You would expect that overexpression of OppZ leads to a decrease in RNA levels from a position within oppB to the end of the operon (except for oppZ, which was overexpressed), but no change in oppA levels.

2) Soften the conclusions regarding CarZ. Evidence for autoregulation is strong, but, the model involving Rho is premature, and should be framed as speculation.

3) Briefly discuss the possibility that the M2 mutation that is intended to prevent OppZ processing may also disrupt base-pairing with oppB.

4) Briefly discuss the differences in time-points used in Figure 1A and TIER-seq. The inactivation of the RNase-TS after 30 min incubation at the nonpermissive temperature shown in Figure 1A does not appear to be very effective given the very modest reduction in 9S processing to 5S rRNA as compared to that observed in *E. coli* (Babitzke and Kushner. PNAS 88:1-5) or more recently in *Salmonella enterica* by Chao et al. (Cell 65: 39-51). Moreover, the results in Figure 1A serve to justify the approach used in the rest of the figure, yet the time points are very different from those chosen for the TIER-seq approach presumably for the reasons mentioned above.

5) Change the title, which is currently too vague. Perhaps something like "Autoregulation of bacterial regulatory RNAs derived from 3' UTRs".

---

## [Author Response]

Essential revision requiring new experimental data:Provide additional experimental support for the model in which OppZ represses translation of oppB, leading to premature Rho termination. Specifically, test whether OppZ represses its own expression when BCM is added. As an example, you could repeat the experiment shown in Figure 4B but with BCM added. It would be useful to look at RNA levels of several opp genes, including oppB and oppZ, and OppB protein levels.

We would like to thank the reviewers for this comment. We have previously attempted to perform the requested experiment, however, we discovered that certain plasmids (including the plasmids carrying a p15a origin of replication used in Figure 4B) are incompatible with BCM treatment. This observation is in line with previous reports showing that Rho is required for plasmid maintenance and suppression of Rho activity causes plasmid-mediated lethality (see PMID: 12946345). To overcome this caveat, we cloned the *oppZregulator* gene onto an established pSC101-based plasmid carrying a *repA* E93K mutation (see PMID: 18295880) that yields a copy number comparable to the previously used p15a plasmids (this origin of replication did not cause lethality upon BCM treatment). We confirmed that OppZ expression from this construct resulted in a significant drop in OppB protein, *oppBCDF* mRNA, and OppZ levels, while the *oppA* mRNA and OppA protein production was constant. Addition of BCM to these samples led to increased *oppBCDF* mRNA and OppZ levels. As expected, OppB protein levels remained repressed due to continued inhibition of translation initiation by OppZ. Again, *oppA* mRNA and OppA did not show significant changes in expression in response to BCM. These new data are shown in Figure 5—figure supplement 1 of our revised manuscript. In addition, we extended the text in the Results section of our manuscript (subsection “OppZ promotes transcription termination through Rho”) to accommodate these new results.

Essential revisions not requiring new experimental data:1) Show the RNA-seq data from Figure 3 for the opp operon. You would expect that overexpression of OppZ leads to a decrease in RNA levels from a position within oppB to the end of the operon (except for oppZ, which was overexpressed), but no change in oppA levels.

As requested by the reviewer, these data have been added to our revised manuscript (new Figure 3—figure supplement 1B). Of note, we did not detect OppZ in the RNA-seq experiment due to a limitation of the NEBNext Ultra II Directional RNA Library Prep Kit, which fails to detect short transcripts such as OppZ.

2) Soften the conclusions regarding CarZ. Evidence for autoregulation is strong, but, the model involving Rho is premature, and should be framed as speculation.

To address this point, we have rephrased wording of the respective paragraph in the Results section. We now write: “Together, these results provide evidence that CarZ is an autoregulatory sRNA and suggest that this function might be more wide-spread among the growing class of 3’ UTRderived sRNAs.” We also changed text in the Abstract and the Introduction section.

3) Briefly discuss the possibility that the M2 mutation that is intended to prevent OppZ processing may also disrupt base-pairing with oppB.

We thank the reviewers for this comment. In our revised manuscript (Figure 3C), we now show that OppZ carrying mutation M2, when expressed from a separate plasmid, efficiently inhibits OppB::GFP production. We also updated the text in the Results section accordingly (subsection “Feedback autoregulation at the suboperonic level”).

4) Briefly discuss the differences in time-points used in Figure 1A and TIER-seq. The inactivation of the RNase-TS after 30 min incubation at the nonpermissive temperature shown in Figure 1A does not appear to be very effective given the very modest reduction in 9S processing to 5S rRNA as compared to that observed in E. coli (Babitzke and Kushner. PNAS 88:1-5) or more recently in Salmonella enterica by Chao et al. (Cell 65: 39-51). Moreover, the results in Figure 1A serve to justify the approach used in the rest of the figure, yet the time points are very different from those chosen for the TIER-seq approach presumably for the reasons mentioned above.

We thank the reviewers for this comment. To address this inconsistency among the experiments, we repeated the experiments shown in Figure 1A and extended the time for RNase E deactivation to 60 minutes. We obtained very similar results and thus updated this figure (and the relevant text in the figure legend) in our revised manuscript.

5) Change the title, which is currently too vague. Perhaps something like "Autoregulation of bacterial regulatory RNAs derived from 3' UTRs".

As requested, we changed the title to: “Gene autoregulation by 3’ UTR-derived small RNAs”.